# Corticothalamic gating of population auditory thalamocortical transmission in mouse

**Baher A Ibrahim[1,2], Caitlin A Murphy[3,4], Georgiy Yudintsev[5], Yoshitaka Shinagawa[1], Matthew I Banks[3,4], Daniel A Llano[1,2,5,6]***

[1]Department of Molecular and Integrative Physiology, University of Illinois, Urbana-Champaign, United States; [2]Beckman Institute for Advanced Science and Technology, University of Illinois, Urbana-Champaign, United States; [3]Department of Neuroscience, School of Medicine and Public Health, University of Wisconsin-Madison, Wisconsin-Madison, United States; [4]Department of Anesthesiology, School of Medicine and Public Health, University of Wisconsin-Madison, Wisconsin-Madison, United States; [5]Neuroscience Program, University of Illinois, Urbana-Champaign, United States; [6]College of Medicine, University of Illinois, Urbana-Champaign, United States

**Abstract** The mechanisms that govern thalamocortical transmission are poorly understood. Recent data have shown that sensory stimuli elicit activity in ensembles of cortical neurons that recapitulate stereotyped spontaneous activity patterns. Here, we elucidate a possible mechanism by which gating of patterned population cortical activity occurs. In this study, sensory-evoked all-or-none cortical population responses were observed in the mouse auditory cortex in vivo and similar stochastic cortical responses were observed in a colliculo-thalamocortical brain slice preparation. Cortical responses were associated with decreases in auditory thalamic synaptic inhibition and increases in thalamic synchrony. Silencing of corticothalamic neurons in layer 6 (but not layer 5) or the thalamic reticular nucleus linearized the cortical responses, suggesting that layer 6 corticothalamic feedback via the thalamic reticular nucleus was responsible for gating stochastic cortical population responses. These data implicate a corticothalamic-thalamic reticular nucleus circuit that modifies thalamic neuronal synchronization to recruit populations of cortical neurons for sensory representations.

*For correspondence:
d-llano@illinois.edu

**Competing interests:** The authors declare that no competing interests exist.

## Introduction

Our experience of the world relies on how sensory information is processed. Thalamocortical projections are critical for activation of the cerebral cortex, which is thought to contain the neural circuits that drive our conscious awareness of sensory stimuli. A classical view of thalamocortical function is that cortical activity during sensory perception is predictable based on activity in the thalamus and patterns of synaptic convergence of thalamocortical axons onto cortical neurons. Thus, sensory processing in this classical scheme behaves as a series of hierarchical linear filters, whose tuning is modified by feedback, creating more complex response properties that reach the level of conscious perception by activating the cortex (*Riesenhuber and Poggio, 1999*; *Vidyasagar and Eysel, 2015*). However, this view does not comport with findings that population activity in sensory cortices is often stereotyped and recapitulates patterns of cortical spontaneous activity (*MacLean et al., 2005*; *Kenet et al., 2003*; *Miller et al., 2014*; *Sakata and Harris, 2009*). As such, a hypothesis has emerged that sensory representations are developed by early exposure to sensory stimuli and stored in the cortex within intracortical networks, and that the thalamus activates these pre-wired

sensory representations upon sensory stimulation (*MacLean et al., 2005*; *Petersen, 2005*). Some observations that support this hypothesis are that spontaneous cortical activity is highly determined by the internal cortical connectivity (*Kenet et al., 2003*; *Sanchez-Vives and McCormick, 2000*), and is not easily predictable from activity in thalamocortical afferents (*MacLean et al., 2005*; *Cossart et al., 2003*). In addition, spontaneous cortical activity is the main substrate for the generation of the internal percepts during states without external sensory input, such as memory recall and dreaming (*Takeuchi et al., 2011*; *Nir and Tononi, 2010*). Further, despite substantial differences in the form and organization in the initial processing stages of different modalities of sensation, at the level of the thalamus and cortex, neural circuits across modalities are relatively homogeneous (*Shepherd, 2011*; *Douglas et al., 1989*; *Phillips et al., 2019*). These findings suggest that there is a common function of thalamocortical circuits that is not tied to specific modalities of perception.

Given that connected neuronal ensembles are likely the main functional unit for behavior and cognition (*Uhlhaas et al., 2009*; *Buzsáki, 2010*), a major control point for the activation of cortical ensembles, and therefore cognition more generally, may lie in the thalamus. Here, we examine the mechanisms by which all-or-none population responses (hereafter called 'Population ON' or 'Population OFF' responses) in the auditory cortex (AC) are gated. We find that gating of cortical activity occurs via corticothalamic projection to the thalamic reticular nucleus (TRN), which is a long-enigmatic structure that partially surrounds and sends GABAergic projections to thalamocortical neurons. Further, we observed that population OFF responses in the AC are associated with increases in synaptic inhibition and neuronal desynchronization in the auditory thalamus. These findings suggest that one mode of thalamocortical function is to select groups of cortical neurons for activation based on feedback from cortical layer 6.

## Results

### Stochastic AC responses to sound presentations in vivo

Transcranial calcium imaging of the AC of an anesthetized GCaMP6s mouse following repeated presentations of a 5 kHz-37 dB SPL pure tone (*Figure 1A*) revealed cortical activity in three distinct areas, the primary AC (A1), secondary AC (A2), and anterior auditory field (AAF), consistent with previous work (*Issa et al., 2014*; *Figure 1B*, MATLAB code in *Figure 1—source code 1*). Such sound-evoked cortical activity represented the average of the cortical responses to 40 presentations of the same tone. Across the 40 trials of the same sound presentation, we observed that A1 showed either a full population response (here called population ON cortical responses) or no response (referred to here as population OFF cortical responses), with some variability within each class (*Figure 1C*, MATLAB code in *Figure 1—source code 1*). Inspection of the individual cortical responses over time revealed the responses to be independent of stimulus presentation order (*Figure 1D*, *Figure 1—source data 1*). Collectively, a histogram of Δf/f of sound-evoked calcium signals from A1 across trials showed two classes of A1 responses ('Population ON' vs. 'Population OFF,' *Figure 1E*, *Figure 1—source data 1*). A greater frequency of the population ON cortical responses was observed with increasing sound pressure level (*Figure 1F*, *Figure 1—source data 1*). Similar stochastic population sensory cortical responses have been observed previously (*MacLean et al., 2005*; *Kenet et al., 2003*; *Miller et al., 2014*; *Sakata and Harris, 2009*), but the mechanisms that determine whether a population ON or OFF response occurs have not yet been elucidated.

### Stochastic AC responses to electrical stimuli to IC in vitro

To examine the circuit mechanisms underlying the stochastic nature of the responses (*Figure 2*), brain slices that retain connectivity between the inferior colliculus (IC), medial geniculate body (MGB), thalamic reticular nucleus (TRN) and AC were constructed (*Figure 2—figure supplement 1*; *Llano et al., 2014*; *Slater et al., 2015*). The auditory colliculo-thalamocortical (aCTC) mouse brain slice retains the synaptic connections between these structures: hence, electrical stimulation of the IC evoked neuronal activity in all of these brain structures as indicated by stimulus-evoked flavoprotein autofluorescence (FA) and calcium-dependent fluorescence signals (*Figure 2A–E*). *Figure 2B and D*, (MATLAB codes in *Figure 2—source code 1* and *2*), show the average of the stimulus-evoked activity of the connected brain structures in the aCTC slice after 10 trials of IC stimulation. However, review of the individual time series of Δf/f of the stimulus-evoked FA and calcium signals

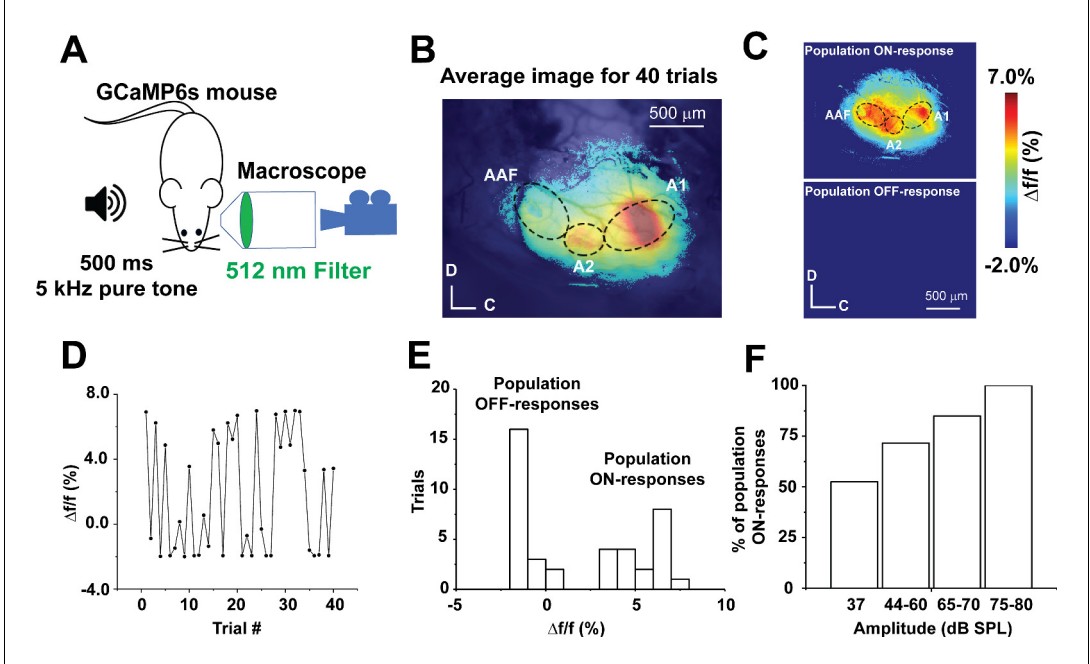

**Figure 1.** Stochastic auditory cortical population responses to repeated sound presentations in vivo. (**A**) A cartoon image showing the experimental design of transcranial calcium imaging of the left AC of GCaMP6s mouse during a 500 ms pure tone (5 kHz) exposure to the right ear. (**B**) Pseudocolor image representing the average map of AC activation indicated by Δf/f of sound-evoked calcium signals following 40 trials of 5 kHz-37 dB SPL pure tone stimulation. (**C**) Two pseudocolor images representing an individual trial for population ON cortical response (top) versus population OFF cortical response (bottom). (**D**) A line graph of Δf/f of the sound-evoked Ca signals of A1 across 40 trials of playing 5 kHz, 37 dB SPL over time. (**E**) A histogram of the Δf/f of sound-evoked calcium signals following 40 trials of 5 kHz-37 dB SPL stimulation (bin size = 1%). (**F**) A bar graph showing the percentage of population ON cortical responses across different sound levels at 5 kHz pure tone; A1: Primary auditory cortex, A2: Secondary auditory cortex, AAF: Anterior auditory field, C: Caudal, D: Dorsal.

The online version of this article includes the following source data and source code for figure 1:

**Source code 1.** The MATLAB code used to produce *Figure 1B and C*.
**Source data 1.** Data values of *Figure 1D-F*.

across trials of IC stimulation (*Figure 2C and E*, *Figure 2—source data 1* and *2*) reveals stochastic responses in the AC; a pattern that was similar to the in vivo data, despite the fact that MGB, TRN, and IC were always responsive to IC stimulation. The FA and calcium signals were determined based on predetermined criteria (*Figure 2—figure supplement 2*, *Figure 2—figure supplement 2— source data 1*, see Materials and methods). This finding was consistent with whole-cell recording from MGB and TRN cells (*Figure 2—figure supplement 3*), where spikes were seen in response to each stimulus irrespective of the presence of a population AC response. We note that similar to previous studies, FA and calcium signals from the MGB are smaller than those from AC (*Dana et al., 2014*; *Llano et al., 2009*), which may be related to limited alignment of mitochondria-containing neuropil processes or low baseline expression of GCaMP in the MGB, respectively.

To ensure that metabolic or imaging artifacts did not drive the observation of these stochastic population cortical responses, local field potentials (LFPs) were recorded in the AC while simultaneously conducting FA imaging, and the LFPs also showed all-or-none response patterns that strongly correlated with the FA signals (*Figure 2F*, *Figure 2—source data 3*). We also observed stochastic population AC responses in brain slices prepared using a different anesthetic (isoflurane) without transcardiac perfusion in a different laboratory using a biphasic stimulator vs. monophasic stimulator (*Figure 2—figure supplement 4*), suggesting that the stochastic responses were not specific to a particular slice preparation or stimulation technique.

Looking more closely at the activity of cortical layer 4 cells, the stimulus-evoked calcium signals of a small population of layer 4 cells following IC stimulation showed that a similar population layer 4 cells was activated with each population ON response and that population OFF responses were

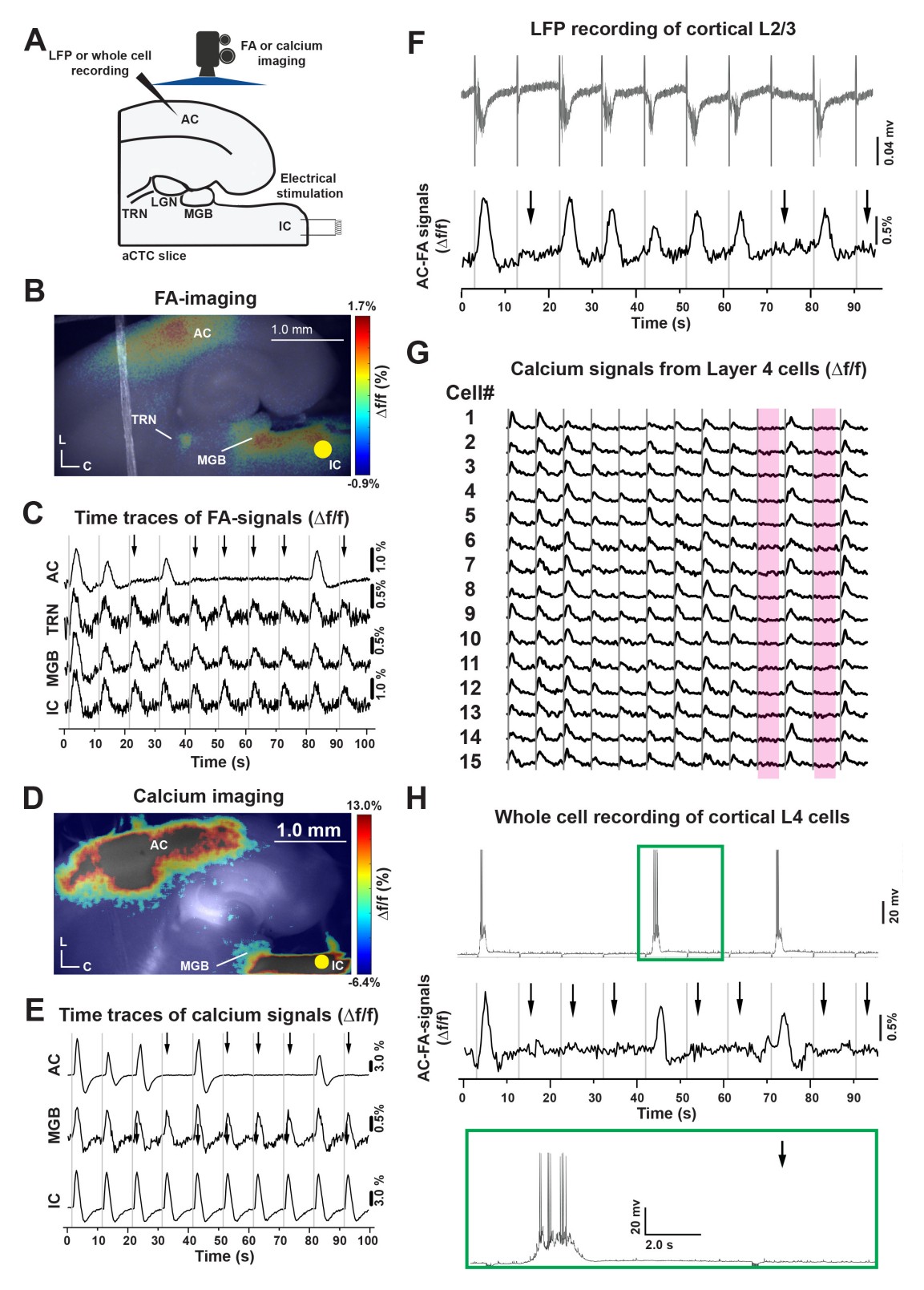

**Figure 2.** Stochastic auditory cortical responses to repeated electrical stimuli in vitro. (**A**) A cartoon image showing the experimental design of simultaneous FA or calcium imaging and IC stimulation of the aCTC slice, which were also associated with LFP or whole-cell recording from cortical 3/4 layers in other experimental setups. (**B and D**) Pseudocolor images showing the neuronal activation indicated by evoked FA or calcium signals, respectively, in the IC, MGB, TRN, and AC following IC stimulation (nine animals). (**C and E**) The time series of Δf/f of evoked FA or calcium signals,

*Figure 2 continued on next page*

*Figure 2 continued*

respectively, in the IC, MGB, TRN, and AC following IC stimulation. (F) The time series of the L3/4 LFP signals (top panel) and Δf/f of the evoked cortical FA signals (bottom panel) following IC stimulation. (G) The time series of evoked calcium signals of a small population of layer 4 cells following IC stimulation, the two rose

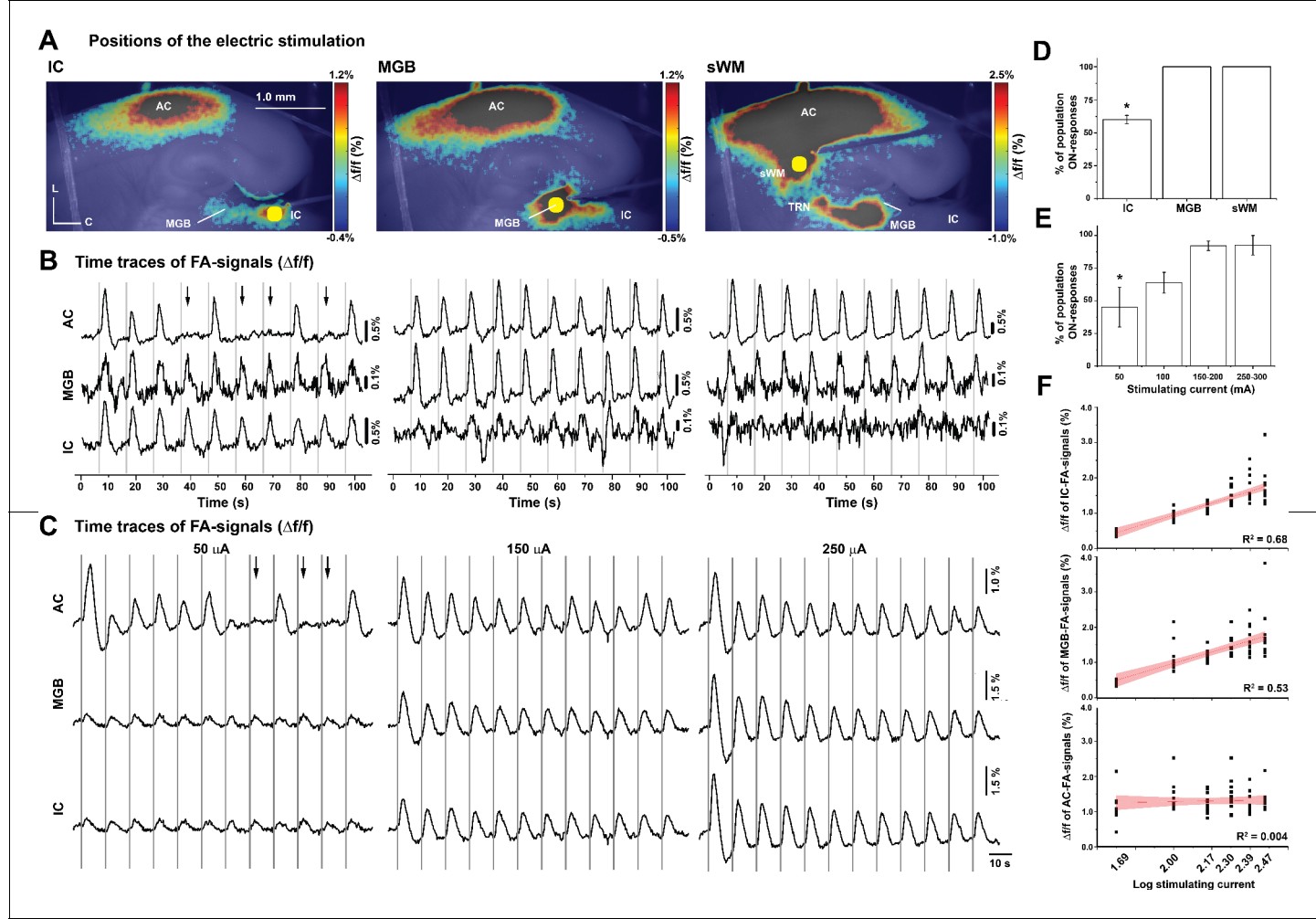

**Figure 3.** The effect of stimulation location and amplitude on the occurrence of OFF-cortical responses. (A) Pseudocolor images of the evoked FA signals in aCTC slice after electrical stimulation of the IC, the MGB, or the subcortical white matter, respectively. (B) The time series of Δf/f of the evoked FA signals in AC, MGB, and IC following electrical stimulation of IC, MGB, or the subcortical white matter, respectively. (C) The time series of Δf/f of the evoked FA signals in AC, MGB, and IC following the stimulation of the IC with different stimulating current amplitudes. (D and E) Bar graphs showing the percentage of population ON cortical responses at different loci of stimulation and across different stimulating current amplitudes at the IC, respectively. (F) Line graphs showing the relationship between stimulating current amplitude and Δf/f of FA signals from IC (top), MGB (middle), and AC (bottom), respectively; Black arrows refer to the missing cortical responses ('population OFF' responses) indicated by the absence of cortical FA, vertical gray lines indicate the onset of IC stimulation, and yellow circles indicate the position of the electrical stimulation; sWM: subcortical white matter.

The online version of this article includes the following source data and source code for figure 3:

**Source code 1.** The main MATLAB code used to produce *Figure 3A*.

**Source code 2.** The supplementary MATLAB code used to produce *Figure 3A*.

**Source data 1.** Time traces of FA signals in *Figure 3B*.

**Source data 2.** Data values of *Figure 3D*.

**Source data 3.** Time traces of FA signals in *Figure 3C*.

**Source data 4.** Data values of *Figure 3E*.

**Source data 5.** Data values of *Figure 3F*.

associated with an absence of layer four responses. For instance, all neurons ceased to respond during stimuli # 9 and 11 (*Figure 2G*, rose ribbons, *Figure 2—source data 4*) and (*Figure 2—figure supplement 5*). Whole-cell recordings of layer 4 cells during population ON population responses revealed a plateau of depolarization upon which rode several action potentials (*Figure 2H*, green box, *Figure 2—source data 5*), resembling previously described UP states (*Sanchez-Vives and McCormick, 2000*; *Rigas and Castro-Alamancos, 2007*; *Fanselow and Connors, 2010*), which was supported by the bimodal distribution shown by the post stimulus membrane potential of such layer 4 cells during population ON cortical responses (*Figure 2—figure supplement 6*, *Figure 2—figure supplement 6—source data 1*, see, Materials and methods). Collectively, these data suggest that IC stimulation produces stochastic population responses in the AC, and that these population responses are stereotyped and represent stimulus-induced UP states.

To determine the locus of control for whether a population ON or population OFF response would be elicited, we moved the stimulating electrode from the IC to the MGB and then to the subcortical white matter (*Figure 3A*, MATLAB codes in *Figure 3—source code 1* and *2*). Population ON/OFF cortical responses occurred only after electrical stimulation of IC, and not by the direct stimulation of MGB or the subcortical white matter (*Figure 3B and D*, *Figure 3—source data 1* and *2*, Kruskal-Wallis: p=0.001, pairwise comparison: p=0.023 for IC vs. MGB or subcortical white matter and p=1 for MGB vs. subcortical white matter, Dunn's post hoc test, n = 5). To determine the impact of stimulating current amplitude on the likelihood of eliciting a population ON-cortical response, different amplitudes of stimulating current were delivered to the IC (*Figure 3C*, *Figure 3—source data 3*). Similar to the in vivo data, the percentage of population ON cortical responses increased with increasing amplitude of the stimulating current (*Figure 3E*, *Figure 3—source data 4*, RM-ANOVA: *$F_{(3,9)}$=9.7, p=0.003, pairwise comparison: *p=0.009 for 50 vs. 150–200 µA, *p=0.008 for 50 vs. 250–300 µA, Bonferroni post hoc test, n = 4 slices from four animals), which was consistent with the in vivo results (*Figure 1F*). Moreover, while there was a log-linear relationship between the FA signals of either IC or MGB and the stimulating current amplitude (*Figure 3F*, *Figure 3—source data 5*, $R^2$ = 0.68, p=2.2×10$^{-19}$ for IC and $R^2$ = 0.53, p=3.3×10$^{-13}$ for MGB), this relationship was lost between the cortical responses and stimulating current amplitude (*Figure 3F*, *Figure 3—source data 5*, $R^2$ = 0.004, p=0.6), suggesting that the magnitude of the cortical response is not predictable from stimulus amplitude.

## Population OFF cortical responses are driven by inhibition in the MGB

Because population OFF cortical responses represented a full absence of stimulus-evoked cortical activity, we reasoned that population OFF cortical responses could be driven by inhibition. To investigate this idea, global disinhibition in the aCTC slice by bath application of gabazine (SR-95531, 200 nM), the GABA$_A$-R blocker, was conducted. Under simultaneous FA imaging and IC stimulation, gabazine perfusion was able to retrieve all missing cortical responses compared to control (*Figure 4B and C*, *Figure 4—source data 1* and *2*, Kruskal-Wallis: p=2.5×10$^{-7}$, pairwise comparison: p=5.08×10$^{-4}$ for gabazine vs. control, p=4.1×10$^{-7}$ for gabazine vs. wash, and p=0.34 for control vs. wash, Dunn's post hoc test, n = 14 trials obtained from five animals), which suggested that the population OFF cortical responses were driven by inhibitory inputs. We further investigated the AC and MGB to search for the site of inhibition that drove population OFF cortical responses. Whole-cell recording of cortical layer 4 or MGB cells voltage-clamped at +10 mV was conducted simultaneously with FA imaging following the stimulation of the IC to track the IPSCs during population ON vs. population OFF cortical responses. Consistent with the previous finding that stimulation of the IC evoked a post-stimulus UP-states activity in layer 4 cells during the population ON cortical responses only (*Figure 2H*; *Fanselow and Connors, 2010*; *Haider et al., 2006*), layer 4 cells demonstrated a surge of evoked post-stimulus IPSCs during population ON cortical responses only (*Figure 4D*). In contrast, MGB cells showed evoked post-stimulus IPSCs following every trial of IC stimulation during population ON and population OFF cortical responses (*Figure 4E*). Quantitatively, while these evoked cortical post-stimulus IPSCs and EPSCS in the layer 4 cells were significantly smaller during population OFF compared to population ON cortical responses (*Figure 4F*, *Figure 4—source data 3*, paired sample Wilcoxon Signed Rank Test: p=0.006 for IPSCs, n = 10 cells from four animals and p=0.036 for EPSCs, n = 6 cells from four animals), the evoked post-stimulus IPSCs in the MGB cells were larger during population OFF compared to population ON cortical responses, with no difference in the net excitatory transferred charges (*Figure 4G*, *Figure 4—source*

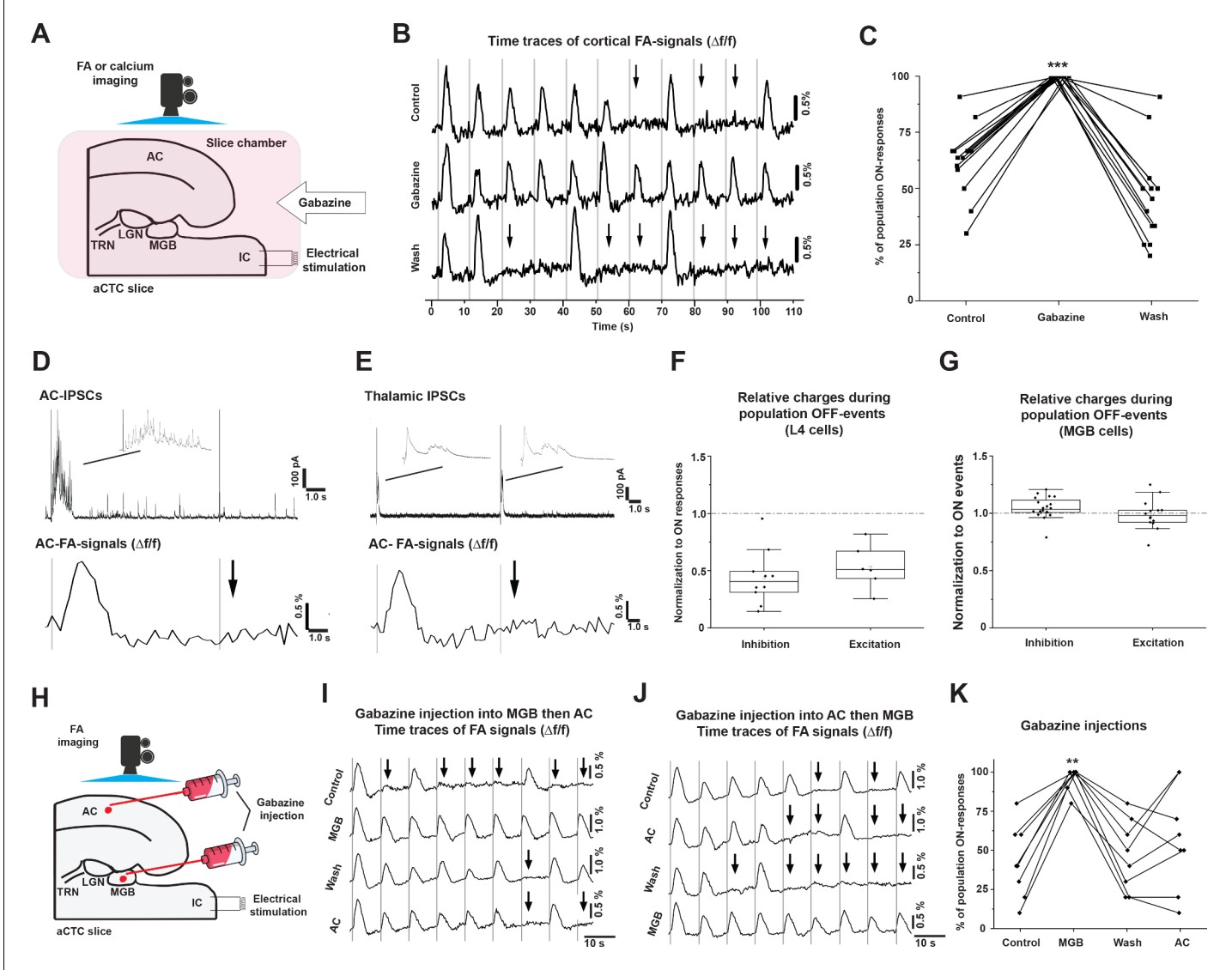

**Figure 4.** The population OFF cortical responses are driven by MGB inhibition. (**A**) A cartoon image showing the experimental design of simultaneous FA imaging and IC stimulation as well as gabazine perfusion. (**B**) The time series of Δf/f of the evoked cortical FA signals with ACSF (control, top trace), with gabazine (middle trace), or wash by ACSF (bottom trace). (**C**) A plot of the results showing that the percentage of population ON cortical responses was significantly higher than that of control and wash. (**D**) Evoked post-stimulus IPSCs recorded from layer 4 cells (top panel), and the Δf/f of the evoked cortical FA responses (bottom panel) following the IC stimulation. (**E**) Evoked post-stimulus IPSCs recorded from MGB cells (top panel), and the Δf/f of the evoked cortical FA responses (bottom panel) following the IC stimulation. (**F and G**) Scatterplots of the area under the curve (AUC) of the evoked post-stimulus IPSCs and EPSCs recorded from AC or MGB, respectively, during population OFF cortical responses and normalized against those recorded during population ON cortical events. (**H**) A cartoon image showing the experimental design of simultaneous FA imaging, IC stimulation, and selective gabazine injection into MGB or AC using a picospritzer. (**I and J**) The time series of Δf/f of the evoked cortical FA signals during a counterbalanced gabazine injection starting into MGB then AC or AC then MGB, respectively. (**K**) A plot of the results showing that the percentage of population ON cortical responses were significantly higher after the injection of gabazine into MGB; Black arrows refer to the occurrence of population OFF cortical responses indicated by the absence of the cortical FA signals; Vertical gray lines indicate the onset of IC stimulation; IPSCs: Inhibitory postsynaptic currents, and EPSCs: Excitatory postsynaptic currents.

The online version of this article includes the following source data and figure supplement(s) for figure 4:

**Source data 1.** Time traces of FA signals in *Figure 4B*.
**Source data 2.** Data values of *Figure 4C*.
**Source data 3.** Data values of *Figure 4F*.
**Source data 4.** Data vaues of *Figure 4G*.
**Source data 5.** Time traces of FA signals in *Figure 4I*.
**Source data 6.** Time traces of FA signals in *Figure 4J*.

*Figure 4 continued on next page*

Figure 4 continued

**Source data 7.** Data values of *Figure 4K*.
**Figure supplement 1.** The injection of gabazine into MGB and AC.

*data 4*, paired sample Wilcoxon Signed Ranks Test: p=0.011 for IPSCs, n = 20 cells from seven animals and p=0.56 for EPSCs, n = 14 cells from four animals). These findings suggest that MGB cells receive more inhibition during population OFF cortical responses with no change in excitation, which led us to hypothesize that MGB activity could be modulated by inhibitory inputs during the population OFF cortical responses.

To test this hypothesis, we determined whether disinhibition of the MGB could yield population ON cortical responses to all stimuli. Under simultaneous FA imaging and IC stimulation, the specific injection of gabazine (*Figure 4—figure supplement 1*) into the MGB in the aCTC slice was able to significantly increase the percentage of population ON cortical responses (*Figure 4I and J*, *Figure 4—source data 5* and *6*), indicated by the Δf/f of the evoked cortical FA signals (*Figure 4K*, *Figure 4—source data 7*, RM-ANOVA: $F_{(3,21)}$ = 13.9, p=3.17×$10^{-5}$, pairwise comparison: *p=5.9×$10^{-5}$, 1.5 × $10^{-4}$, 0.0026 for gabazine in MGB vs. control, wash, and gabazine in AC, respectively, Bonferroni post hoc test, n = 8 slices from seven animals). In contrast, the same effect was not observed after selective gabazine injection into the AC (*Figure 4K*, *Figure 4—source data 7*, RM-ANOVA: $F_{(3,21)}$ = 13.9, pairwise comparison: p=0.73 and 1.0 for gabazine in AC vs. control and wash, respectively, Bonferroni post hoc test, n = 8 slices from seven animals), consistent with data obtained from the whole-cell recording of layer 4 cells (*Figure 4D and F*). Accordingly, these data confirmed that the population OFF cortical responses could be driven by thalamic inhibition.

## Synchronized MGB cells are associated with population ON-cortical responses

Although the mean peak latencies of IPSCs and EPSCs received by MGB cells showed no difference during population ON and population OFF cortical responses (*Figure 5A and B*, *Figure 5—source data 1*, paired sample Wilcoxon Signed Ranks Test: p=0.28, n = 20 cells from seven animals for IPSCs and paired t-test: $t_{(12)}$ = −0.72, p=0.48, n = 14 cells from four animals for EPSCs), the cumulative distribution function showed that some MGB cells (~50% of the recorded cells) received earlier IPSCs during population OFF cortical responses (*Figure 5C*, *Figure 5—source data 1*). These earlier inhibitory signals received by some MGB cells could negatively impact the synchronization between the MGB cells required to pass the sensory information to the cortex. To test this hypothesis, the time courses of calcium signals of MGB cells, which were found to match the voltage signals that were simultaneously recorded in a separate experiment (*Figure 5—figure supplement 1*), were imaged following the stimulation of the IC, and were compared during population ON vs. population OFF cortical responses (*Figure 5D and E*, *Figure 5—source data 2*). We observed that the variance of the peak latencies of the evoked calcium signals from all thalamic cells was larger during population OFF cortical responses than population ON responses (*Figure 5F*, *Figure 5—source data 3*, paired sample t-test: $t_{(13)}$ = −2.37, *p=0.033, n = 14 paired responses (population ON vs. population OFF) including 339 cells from five animals). We note that the absolute latencies of the calcium imaging responses are considerably longer than those measured electrophysiologically. However, despite the relatively slow time course of the calcium response, the peak times remained relatively stable during population ON responses. These data suggest that synchronous thalamic relay cell activity is required to evoke a cortical population ON response, consistent with previous work (*Bruno and Sakmann, 2006*).

## Corticothalamic layer 6 cells are the main driver of the missing cortical response via TRN

The data described above suggest that MGB cells receive more inhibition during population OFF cortical responses. Therefore, we investigated the source of these potential inhibitory inputs. As previously reported, the MGB can be inhibited by the cortex through the feedback inhibition by TRN (*Lam and Sherman, 2010*) or by the IC through the feedforward inhibition by IC GABAergic cells (*Peruzzi et al., 1997*; *Winer et al., 1996*). Therefore, we determined if inhibition of either of these

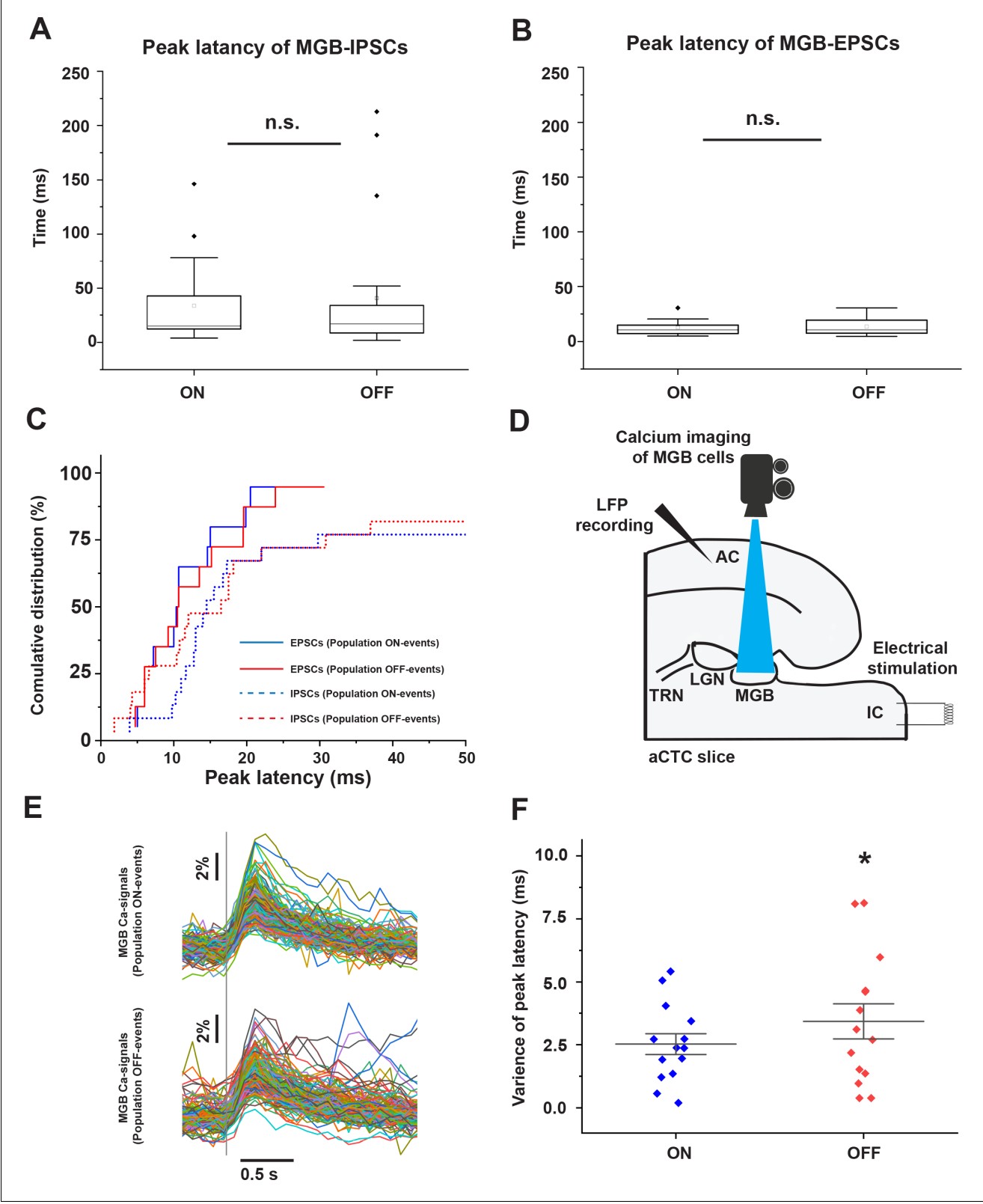

**Figure 5.** Desynchronized MGB cellular activity is associated with population OFF-cortical responses. (A and B) Plots of peak latencies of IPSCs (A) and EPSCs (B) measured in the MGB after IC stimulation. No difference was seen in the mean latency during population ON vs. OFF responses. (C) Cumulative distribution functions of the peak latencies of IPSCs (dotted line) and EPSCs (solid line) during population ON (blue line) or population OFF-cortical responses (red line) showing that ~50% of MGB cells received earlier IPSCs during population OFF cortical responses (red dotted line). (D) A
*Figure 5 continued on next page*

*Figure 5 continued*

cartoon image showing the experimental design of simultaneous calcium imaging of MGB, LFP recording form the cortex, and the IC stimulation. (**E**) The sweeps of the evoked calcium signals of all activated cells during population ON (top) vs. population OFF (bottom) cortical responses indicated by the post-stimulus cortical LFP signals recorded from L3/4. (**F**) A scatterplot showing a higher variance of the peak latencies of MGB cells activated during population OFF cortical response compared to those activated during population ON cortical responses. Vertical gray line indicates the onset of IC stimulation.

The online version of this article includes the following source data and figure supplement(s) for figure 5:

**Source data 1.** Data values of *Figure 5A-C*.
**Source data 2.** Time traces of calcium signals in *Figure 5E*.
**Source data 3.** Data values of *Figure 5F*.
**Figure supplement 1.** Calcium imaging from MGB cells.

---

pathways (to disinhibit the MGB) could retrieve the missing cortical responses. Corticothalamic layer 6 cells indirectly inhibit thalamic relay cells by exciting GABAergic neurons in the TRN, a shell-like structure of GABAergic neurons that surrounds the most of dorsolateral part of the thalamus (*Crandall et al., 2015*; *Guo et al., 2017*). Therefore, the feedback inhibition of MGB via corticothalamic layer 6-TRN pathway was examined. Injection of the AC of *NTSR1*-Cre neonates with halorhodopsin-AAV resulted in a successful Cre-dependent expression of eNpHR3.0 receptors in the corticothalamic layer 6 as well as their projections to TRN and MGB indicated by YFP tag (*Figure 6B*). Photoinhibition of corticothalamic layer 6 cells by illumination with 565 nm light resulted in a significant increase in the probability of population ON cortical responses as indicated by the recovery of the post-stimulus LFP signals from L3/4 compared to the control (*Figure 6C and D*, *Figure 6—source data 1*, paired t-test: $t_{(15)} = -7.06$, **p=$3.8\times10^{-6}$, n = 16 trials from four animals). A control experiment was done by injecting *NTSR1*-Cre mice with a virus expressing YFP without the inhibitory opsin. The same illumination of the aCTC slice expressing YFP in layer 6 cells (*Figure 6F*) with a 565 nm light of could not retrieve the population ON cortical responses (*Figure 6G and H*, *Figure 6—source data 2*, paired t-test: $t_{(24)} = -1.3$, p=0.2, n = 25 trials from five animals). To examine the impact of silencing corticothalamic layer 6 neurons using FA imaging, the AC of a separate group of *NTSR1*-Cre neonates was injected with DREADDs-AAV virus, which resulted in a successful Cre-dependent expression of the inhibitory chemogenetic receptors, hM4Di, as indicated by mCherry tag specifically in corticothalamic layer 6 cells as well as their projections to TRN and MGB (*Figure 6—figure supplement 1B*). Consistent with the previous data, the chemical inhibition of corticothalamic layer 6 cells expressing hM4Di receptors by their chemical actuator, CNO, significantly increased the frequency of population ON cortical responses as indicated by Δf/f of evoked cortical FA signals compared to the control (*Figure 6—figure supplement 1C and D*, *Figure 6—figure supplement 1—source data 1* and *2*, paired test: $t_{(3)} = -3.66$, p=0.035, n = 4 trials from four animals). The control experiment was done by injecting other *NTSR1*-Cre mice with a virus expressing mCherry without the DREADDs, and the perfusion of CNO to the aCTC slices taken from these animals could not retrieve the population ON cortical responses (*Figure 6—figure supplement 1E*, *Figure 6—figure supplement 1—source data 3*, paired test: $t_{(4)} = 0.9$, p=0.4, n = 5 trials from five animals).

Given that corticothalamic layer 6 cells project to MGB through a direct excitatory synapse, further examination was required to test if the TRN is the main driver of corticothalamic layer six effect. Blocking of the TRN activity by NBQX, the AMPA-R blocker (*Sun et al., 2013*; *Lee et al., 2010a*), which was specifically injected into TRN, significantly increased the probability of population ON cortical events indicated by Δf/f of the evoked cortical FA signals (*Figure 6—figure supplement 1G and H*, *Figure 6—figure supplement 1—source data 4* and *5*, paired test: $t_{(5)} = -6.03$, **p=$5.2\times10^{-4}$, n = 8 trials from seven animals). To test the specificity of corticothalamic layer 6-TRN pathway to modulate MGB activity, we performed a control experiment by inhibiting layer 5 corticothalamic cells, which do not have significant projections to the TRN (*Crabtree, 2018*). The injection of the AC of *RBP4*-Cre neonatal mice with halorhodopsin virus resulted in a successful Cre-dependent expression of eNpHR3.0 receptors in layer 5 cells as indicated by YFP tag (*Figure 6J*). In contrast to layer 6 cells, the projections from layer five to MGB in the aCTC slice are sparse, as has been previously described (*Llano and Sherman, 2008*), so they are not well seen in the low-magnification image (*Figure 6J*). However, with high magnification, the layer 5 projections to MGB were observed

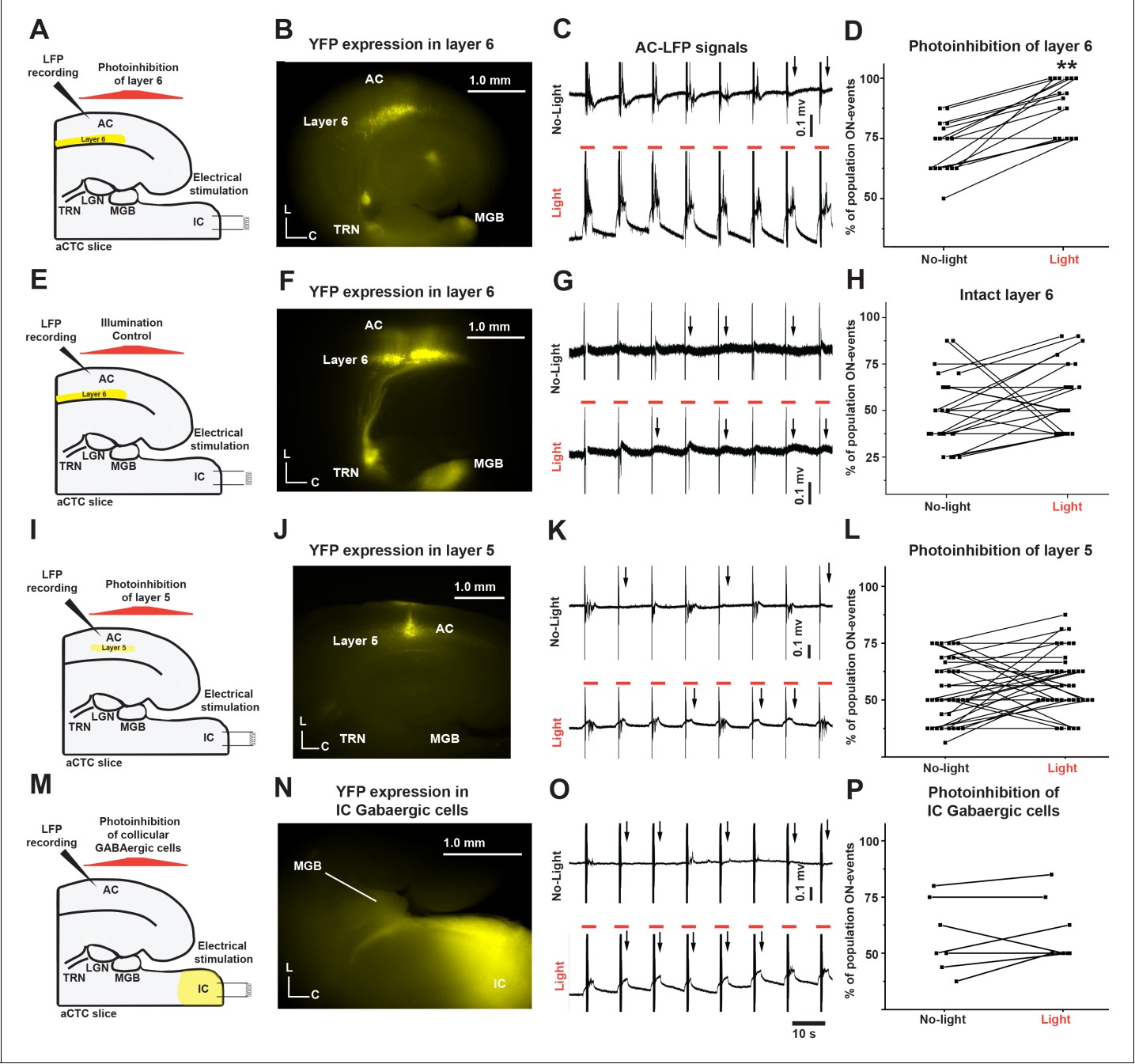

**Figure 6.** The population OFF cortical responses were driven by the feedback inhibition of MGB by corticothalamic layer six via TRN. (**A and E**) Cartoon images showing the experimental design of simultaneous IC stimulation, LFP recording, and photoinhibition for aCTC slice with and without eNpHR3.0 receptors, respectively. (**B and F**) Images of aCTC slice of *NTSR1*-Cre mouse showing the expression of eNpHR3.0 receptors as indicated by YFP tag and YFP only, respectively, in *NTSR1*-positive corticothalamic layer 6 cells as well as their projections to TRN and MGB. (**C, G, K and O**) The time series of the post-stimulus cortical LFP signals from L3/4 following IC stimulation without (top panel) and with 565 nm light (bottom panel). (**D**) A scatterplot of percent population ON responses pre- and post-light application showing that the percentage of population ON cortical events was higher during the photoinhibition of corticothalamic layer 6 cells. (**H**) A scatterplot of percent population ON responses pre- and post-light application showing that the illumination of the aCTC slice expressing no inhibitory opsin with 565 nm light could not retrieve the population ON cortical responses. (**I**) A cartoon image showing the experimental design of simultaneous IC stimulation, LFP recording, and photoinhibition of layer 5 cells (**J**) Image of aCTC slice of *RBP4*-Cre mouse showing the expression of eNpHR3.0 receptors as indicated by YFP tag in RBP4-positive layer 5 cells. (**L**) A scatterplot of percent population ON responses pre- and post-light application showing that the photoinhibition of layer 5 cells could not retrieve the population ON cortical responses. (**M**) A cartoon image showing the experimental design of simultaneous IC stimulation, LFP recording, and photoinhibition of IC GABAergic cells (**N**) Image of aCTC slice from *GAD2*-Cre mouse showing the Cre-dependent expression of halorhodopsin indicated by YFP tag in GABAergic cells

*Figure 6 continued on next page*

*Figure 6 continued*

of the IC as well as their projections to MGB. (P) A scatterplot of the percentage of cortical population ON responses showing no change after the photoinhibition of IC GABAergic cells by light; Black arrows refer to the occurrence of population OFF cortical responses indicated by the absence of post-stimulus cortical LFP signals from L3/4; Vertical gray lines indicate the onset of IC stimulation; Orange lines indicate the time period of illumination (3 s).

The online version of this article includes the following source data and figure supplement(s) for figure 6:

**Source data 1.** Data values of *Figure 6D*.
**Source data 2.** Data values of *Figure 6H*.
**Source data 3.** Data values of *Figure 6L*.
**Source data 4.** Data values of *Figure 6P*.
**Figure supplement 1.** Chemical inhibition of corticothalamic layer 6 cells and TRN blocking retrieved the missing cortical responses.
**Figure supplement 1—source data 1.** Time traces of FA signals in *Figure 6—figure supplement 1C*.
**Figure supplement 1—source data 2.** Data values of *Figure 6—figure supplement 1D*.
**Figure supplement 1—source data 3.** Data values of *Figure 6—figure supplement 1E*.
**Figure supplement 1—source data 4.** Time traces of FA signals in *Figure 6—figure supplement 1G*.
**Figure supplement 1—source data 5.** Data values of *Figure 6—figure supplement 1H*.
**Figure supplement 2.** The projections from layer 5 cells to the MGB in the aCTC slice.

(*Figure 6—figure supplement 2*). Consistent with the previous data, the photoinhibition of layer 5 cells expressing eNpHR3.0 receptors by illumination with 565 nm light could not retrieve the missing cortical responses, indicated by no recovery of the post-stimulus LFP signals from L3/4 compared to control (no light) (*Figure 6K and L*, *Figure 6—source data 3*, paired t-test: t(36) = 0.46, p=0.18, n = 37 trials from four animals), which suggests that the population OFF cortical responses were specifically driven by MGB inhibition through corticothalamic layer 6-TRN pathway. To suppress feedforward inhibition, the IC of neonatal *GAD2*-Cre mice was injected with halorhodopsin-AAV virus (see Materials and methods) to induce the expression of halorhodopsin specifically in the GABAergic cells of the IC in a Cre-dependent manner. As expected, GABAergic cells of the IC as well as their projections to MGB expressed halorhodopsin indicated by the presence of YFP (*Figure 6N*). Photoinhibition of GABAergic cells of the IC by illumination with 565 nm light was not able to retrieve the missing cortical responses as indicated by no recovery of the post-stimulus LFP signals recorded from L3/4 (*Figure 6O and P*, *Figure 6—source data 4*, Paired t-test: t(6) = −1.03, p=0.34, n = 7 trials from five animals). Accordingly, based on the data shown in *Figure 6* and *Figure 6—figure supplement 1*, we conclude that the population OFF cortical responses were most likely driven by the feedback inhibition of MGB by corticothalamic layer 6 cells via TRN.

## Discussion

We observed stochastic population cortical responses in the mouse AC following the presentations of pure tones in vivo or electrical stimulation of the IC in vitro. Population ON responses were associated with synchronized responses among MGB cells, whereas population OFF responses were associated with TRN-mediated inhibition at the level of the MGB, under the control of layer 6 corticothalamic projections. Other inhibitory projections to the MGB from the IC had no impact on the probability of eliciting a population ON response. It is unlikely that population OFF cortical responses were a sign of cortical adaptation, because adaptation responses are generally characterized by gradual decrease of the response amplitude, rather than the all-or-none responses observed here (*Chung et al., 2002*; *Abolafia et al., 2011*). We conclude from these findings that layer 6 corticothalamic neurons gate population activity in the AC via their projections to TRN, which desynchronize MGB neurons. Previous work has suggested that corticothalamic axons, via the strong inhibition to thalamocortical cells by way of the TRN (*Crandall et al., 2015*; *Olsen et al., 2012*; *Paz and Huguenard, 2015*; *Steriade et al., 1996*; *Guillery and Harting, 2003*; *McAlonan et al., 2008*), control thalamocortical information flow (*Destexhe, 2000*; *Yu et al., 2004*). In the current study, the gating mechanism appears to involve TRN-based desynchronization of thalamocortical neurons rather than diminishing the overall thalamic response, which is consistent with previous findings showing that the TRN can desynchronize thalamocortical cells via multiple mechanisms (*Pita-Almenar et al., 2014*). This explanation is also consistent with our finding that direct electrical

stimulation of the thalamus did not lead to stochastic AC response because direct electrical stimulation is likely to elicit highly synchronous responses among MGB neurons. It is also possible that silencing layer six corticothalamic neurons led to changes in likelihood of activating the AC via intracortical connections of *NTSR1*-positive neurons. Future work involving selective silencing of cortico-reticular terminals vs. corticocortical terminals will be needed to distinguish between these possibilities. It will also be important to determine if the mechanisms observed in the current slice data are also seen in adult animals. Juveniles were used to maximize slice connectivity, but now that specific hypotheses may be proposed, elements of the proposed thalamocortical gating mechanisms can be examined in adult preparations. We also note that population OFF cortical responses in the current study were never initiated by the first stimulation of the IC. Rather they were always elicited by the subsequent IC stimulation, which suggests that population OFF cortical responses occurred only after evoked cortical activity that could change the internal dynamics of the cortical cells. Moreover, the strong association between the high probability of the OFF cortical responses and the low level of stimulation suggests that cortico-reticulothalamic control of thalamocortical transmission is most likely to be effective when signal-to-noise ratio is low. This finding is consistent with the notion that top-down modulation is mostly required for the attentional modulation of weak signals and that highly salient signals rely on bottom-up mechanisms to activate perceptual representations (reviewed in *Asilador and Llano, 2020*). Classical models have viewed sensory processing stations, including the thalamus and cortex, as a series of feedforward, hierarchically organized filters whereby combinations of receptive fields produce increasingly selective feature detectors, culminating in uniquely selective neurons (*Riesenhuber and Poggio, 1999*; *Vidyasagar and Eysel, 2015*). This type of organization implies that moment-to-moment perception is a reflection of detailed streams of information coursing through ascending sensory systems, to be consciously perceived when those streams engage highly selective cells in the cortex. An alternative view is that ascending information is used to create and modify a bank of sensory representations that are recruited depending on behavioral needs, and that conscious perception reflects activation of these pre-wired circuits (*Ringach, 2009*; *Llinás, 1994*; *Mishkin, 1982*). Thus, conscious perception may involve the release of stereotyped patterns of cortical activity. One potential role of thalamocortical transmission in this model is to select cortical ensembles rather than impress sensory information upon them. Our data suggest that at the thalamocortical level, sensory inputs activate stereotyped patterns of population cortical activity. Future work in awake preparations will help to establish whether population ON responses correspond to perception of sensory stimuli.

Critical for any such mechanism of control of cortical ensembles is a means by which those ensembles are selected. The current data suggest that the TRN, under the control of layer 6 projections, gates populations of thalamic neurons by desynchronizing their responses, decreasing the likelihood of engendering a population cortical response. A mechanism of TRN-based modification of thalamic synchrony to activate the cortex has been proposed previously (*Saalmann and Kastner, 2011*) and is consistent with the finding that synchronized populations of thalamic neurons are required to optimally activate the cortex (*Bruno and Sakmann, 2006*) and that the TRN is at the heart of a prefrontal cortex-based mechanism to shape cortical activation under changing cognitive demands (*Zikopoulos and Barbas, 2006*; *Wimmer et al., 2015*; *Yingling and Skinner, 1975*). Further, the TRN receives inputs from basal forebrain, amygdala and non-reciprocally linked regions of the thalamus (*Kimura et al., 2007*; *Lam and Sherman, 2005*; *Lam and Sherman, 2015*; *Crabtree et al., 1998*; *Crabtree and Isaac, 2002*; *Lee et al., 2010b*; *Desîlets-Roy et al., 2002*; *Zikopoulos and Barbas, 2012*; *Aizenberg et al., 2019*), forming an assortment of inputs to potentially modulate TRN, and ultimately select cortical circuits for activation. Given the putative role of the TRN in the selection of thalamic, and therefore cortical circuits during sensory perception, one would predict that disruption of TRN activity could lead to uncontrolled release of patterns of cortical activity. Consistent with this idea, ample evidence has accumulated to suggest that schizophrenia, a disease characterized by the presence of auditory hallucinations, involves disruption of the TRN (*Pratt and Morris, 2015*; *Ferrarelli and Tononi, 2011*; *Young and Wimmer, 2017*; *Bencherif et al., 2012*; *Light and Braff, 1999*; *Patterson et al., 2008*; *Ferrarelli et al., 2007*; *Ferrarelli et al., 2010*; *Wamsley et al., 2012*; *Ferrarelli and Tononi, 2017*).

## Conclusion

Here, we describe a unique stimulus-evoked population cortical all-or-none response, which suggests that thalamus recruits cortical ensembles of pre-wired sensory representations upon external stimulation in conjunction with internal cortical dynamics of corticothalamic neurons. These data also suggest that corticothalamic modulators control the temporal coordination between the thalamic cells to gate the activation of the intracortical network. It will be important in future studies to more fully understand how other regulators of the TRN, such as the basal forebrain, prefrontal cortex and amygdala, influence the selection of cortical circuits during active behavior.

# Materials and methods

## Animals

C57BL/6J (Jackson Laboratory, stock # 000664), C57BL/6J-Tg (*Thy1*-GCaMP6s) GP4.3Dkim/J a.k.a. GCaMP6s mice (Jackson Laboratory, stock # 024275), BALB/c (Jackson Laboratory, stock # 000651), *Gad2*-IRES-Cre (Jackson Laboratory, stock # 010802), *NTSR1*-Cre (MMRRC, 017266-UCD) and *RBP4*-Cre (MMRRC, 031125-UCD). Mice of both sexes were used. All applicable guidelines for the care and use of animals were followed. All surgical procedures were approved by the Institutional Animal Care and Use Committee (IACUC). Animals were housed in animal care facilities approved by the Association for Assessment and Accreditation of Laboratory Animal Care (AAALAC).

## In vivo imaging

The detailed procedures have been described before (*Yudintsev et al., 2020*). In brief, adult GCaMP6s mice were used for transcranial in vivo imaging of evoked calcium signals from the left AC. For each experiment, the mouse was anesthetized with a mixture of ketamine and xylazine (100 mg/kg and 3 mg/kg, respectively) delivered intraperitoneally. The animal's body temperature was maintained within the range of 35.5°C and 37°C using a DC temperature controller (FHC, ME, USA). Mid-sagittal and mid-lateral incisions were made to expose the dorsal and lateral aspects of the skull along with the temporalis muscle. The temporalis muscle was separated from the skull to expose the ventral parts of the underlying AC. The site was cleaned with sterile saline, and the surface of the skull was thinned by a surgical drill. A small amount of dental cement (3M ESPE KETAC) was mixed to a medium level of viscosity and added to the head of the bolt just enough to cover it. A head-bolt was bonded to the top of the skull, and the dental cement was allowed to set. An Imager 3001 integrated data acquisition and analysis system (Optical Imaging Ltd., Israel) was used to image the cortical responses to sound in mice. A macroscope consisting of 85 mm f/1.4 and 50 mm f/1.2 Nikon lenses was mounted to an Adimec 1000 m high-end CCD camera (7.4 × 7.4 µm pixel size, 1004 × 1004 resolution), and centered above the left AC, focused approximately 0.5 mm below the surface of the exposed skull. Acoustic stimuli were generated using a TDT system three with an RP 2.1 enhanced real-time processor and delivered via an ES1 free field electrostatic speaker (Tucker-Davis Technologies, FL, USA), located approximately 8 cm away from the contralateral ear. All imaging experiments were conducted in a sound-proof chamber and images were obtained at 10 frames per second. Each trial of sound presentation was composed of two conditions (10 s each); Condition 0 (C0), where there is no sound and condition 1 (C1), where there is a 500 ms sound presentation that comes after 5 s from the onset of the C1. For imaging, the blue light exposure was only on during the 10 s of each trial, and there was a 5 s interval between the two conditions during which the blue light was off. This intermittent schedule of the blue light exposure was done to lower the likelihood of photobleaching. 500 ms pure tones of 5 kHz, at 37, 44, 50, 55, 60, 65, 70, 75, or 80 dB SPL were used for acoustic stimulation. The response window was set as one second starting from the $5^{th}$ second for C0 or from the onset of sound for C1. The Δf/f of the response window was computed as the difference between $\Delta f/f_0$ (C0) and $\Delta f/f_1$ (C1) using a custom MATLAB code (*Figure 1—source code 1*).

## Virus injection

To modulate specific cell types, Cre recombinase-expressing mice were used to provide an expression of opto- or chemo-genetic probes in those specific cells 11 days after viral injection at P4. The detailed procedures have been described before (*Huynh et al., 2020*). For all neonates,

cryoanesthesia was induced after five to ten minutes. A toe pinch was done to confirm that the mice were fully anesthetized. A small animal stereotaxic instrument (David Kopf Instruments, Tujunga, CA) was used with a universal syringe holder (David Kopf Instruments, Tujunga, CA) and standard ear bars with rubber tips (Stoelting, Wood Dale, IL) were used to stabilize the head. The adaptor stage was cooled by adding ethanol and dry ice to an attached well. A temperature label (RLC-60-26/56, Omega, Norwalk, CT) was attached to the stage to provide the temperature of the stage during cooling. The temperature was kept above 2°C to prevent hypothermia or cold-induced skin damage of the neonatal mice and below 8°C to sustain cryoanesthesia. Glass micropipettes (3.5-inches, World Precision Instruments, Sarasota, FL) were pulled using a micropipette puller (P-97, Sutter Instruments, Novato, CA) and broken back to a tip diameter between 35 and 50 µm.

The micropipette was filled with mineral oil (Thermo Fisher Scientific Inc, Waltham, MA) and attached to a pressure injector (Nanoliter 2010, World Precision Instruments, Sarasota, FL) connected to a pump controller (Micro4 Controller, World Precision Instruments, Sarasota, FL). The AC of *NTSR1*-Cre (*Olsen et al., 2012*; *Bortone et al., 2014*; *Kim et al., 2014*; *Mease et al., 2014*) or *RBP4*-Cre (*Jeong et al., 2016*; *Grant et al., 2016*) neonates was injected with eNpHR3.0 AAV1 (AAV-EF1a-DIO-eNpHR3.0-YFP with titer equal to $4.7–5.7 \times 10^{12}$, referred to hereafter as halorhodopsin-AAV) constructs from UNC Vector Core (Chapel Hill, NC), AAV1-Ef1a-DIO EYFP (control for halorhodopsin), Gi-coupled hM4Di DREADDs AAV8 (AAV8-DIO-hSyn-hM4Di-mCherry, referred to here as DREADDs-AAV), AAV8-hSyn-DIO-mCherry (control for DREADDs), or AAV9-FLEX-EGFP (for the histology done on *RBP4*-Cre mouse) constructs from Addgene (Cambridge, MA). The micropipette carrying the viral particles was first located above the AC in the left hemisphere at 1.5 mm anterior to lambda and just at the edge of the skull's flat horizon. The tip was lowered to 1.2 mm from the brain surface and was then pulled back to 1.0 mm for the first injection where 200 nL of halorhodopsin-AAV or DREADDs-AAV was injected at 200 nL/min. After the injection was finished, the micropipette was left in the brain for 1 min before removing to allow the injectate to settle into the brain. Following the first injection, the tip was pulled back stepwise in 0.1 mm increments, and 200 nL of the injectate was injected at every step until the tip reached 0.3 mm from the surface. In total, 1600 nL of AAV was injected into the AC. The incision was sutured using 5/0 thread size, nylon sutures (CP Medical, Norcross, GA). To target the GABAergic cells of the inferior colliculus (IC), the IC of *GAD2*-Cre (*Taniguchi et al., 2011*; *Lammel et al., 2015*; *Villalobos et al., 2018*) neonatal mice was injected with halorhodopsin-AAV following the same procedures shown above, but the micropipette loaded by halorhodopsin-AAV was located over the IC at the left hemisphere at 2.0 mm posterior to lambda and 1.0 mm laterally from the midline. The neonates were transferred back onto a warming pad to recover. After 5–7 min, their skin color was returned to normal, and they started moving. After recovery, all neonates were returned to their nest with the parents.

## Brain slicing

For all in vitro experiments, 15- to 18-day-old mice were initially anesthetized with ketamine (100 mg/ kg) and xylazine (3 mg/kg) intraperitoneally and transcardially perfused with chilled (4°C) sucrose-based slicing solution containing the following (in mM): 234 sucrose, 11 glucose, 26 NaHCO$_3$, 2.5 KCl, 1.25 NaH$_2$PO$_4$, 10 MgCl$_2$, 0.5 CaCl$_2$. After the brain was removed from the skull, it was cut to obtain auditory colliculo-thalamocortical brain slice (aCTC) as shown (*Figure 2—figure supplement 1*) and as described before (*Llano et al., 2014*; *Slater et al., 2015*). 600 µm thick horizontal brain slices were obtained to retain the connectivity between IC, MGB, TRN and AC. All slices were incubated for 30 min at 33°C in a solution composed of (in mM: 26 NaHCO$_3$, 2.5 KCl, 10 glucose, 126 NaCl, 1.25 NaH$_2$PO$_4$, 3 MgCl$_2$, and 1 CaCl$_2$). After incubation, all slices were transferred to a perfusion chamber coupled to an upright Olympus BX51 microscope, perfused with artificial cerebrospinal fluid (ACSF) containing (in mM) 26 NaHCO$_3$, 2.5 KCl, 10 glucose, 126 NaCl, 1.25 NaH$_2$PO$_4$, 2 MgCl$_2$, and 2 CaCl$_2$. Another set of experiments was done in a different laboratory to exclude any experimental factors related to our laboratory environment, chemicals, or anesthesia. As reported previously (*Krause et al., 2014*), following full anesthesia by isoflurane, a C57BL/6J mouse was immediately decapitated without cardiac perfusion, the animal's brain was extracted and immersed in cutting artificial CSF [cACSF; composed of (in mM) 111 NaCl, 35 NaHCO$_3$, 20 HEPES, 1.8 KCl, 1.05 CaCl$_2$, 2.8 MgSO$_4$, 1.2 KH$_2$PO$_4$, and 10 glucose] at 0–4°C. Slices were maintained in cACSF at 24°C for >1 hr before transfer to the recording chamber, which was perfused at 3–6 ml/min with ACSF [composed of (in mM) 111 NaCl, 35 NaHCO$_3$, 20 HEPES, 1.8 KCl, 2.1 CaCl$_2$, 1.4

MgSO$_4$, 1.2 KH2PO$_4$, and 10 glucose]. All of the solutions were bubbled with 95% oxygen/5% carbon dioxide and all experiments were done at room temperature.

## Electrical stimulation

All the electrical stimulation protocols in the IC evoked similar cortical response patterns. One second electrical train pulses of (250 µA, 40 Hz, 1 ms pulse width) to IC was the main stimulating protocol as described before (*Ibrahim et al., 2017*). However, one second durations of electrical stimulation were not suitable for electrophysiology experiments because responses were intermingled with the stimulus artifact. Therefore, for whole-cell and LFP recording, the stimulation of IC was done with a single 3 ms pulse (300–500 µA) or 100 ms long trains of pulses (250 µA, 40 Hz, 1 ms pulse width) were used. The electrical stimulation was done using a concentric bipolar electrode (Cat#30201, FHC) every 10–20 s. The parameters of the electrical pulses were adjusted by a B and K precision wave generator (model # 4063) and World Precision Instruments stimulation isolator (A-360). In another stimulation set up (Our collaborator's laboratory), a biphasic current pulse (200 µA, 5 ms; STG4002 stimulator, Multichannel Systems, Reutlingen, Germany) was delivered to IC at 0.05 Hz using bipolar tungsten electrodes (100 KΩ, FHC Inc, Bowdoin, ME).

## Imaging

For calcium imaging in vitro, slices from GCaMP6s mice or wild-type mice loaded with CAL-520, AM (Abcam, ab171868) calcium dye were used. For CAL-520, AM calcium dye loading, the aCTC slices were incubated in 48 µl of DMSO dye solution +2 µl of Pluronic F-127 (Cat# P6866, Invitrogen) in 2.5 ml of the incubating solution at 35–36°C for 25–28 min according to *MacLean and Yuste, 2009*; *Yuste et al., 2011*. The slices then were incubated in normal incubating solution (shown above) for 30 min to wash the extra dye. Imaging was done under ACSF perfusion as described before. Depending on the experiment, the evoked calcium or flavoprotein autofluorescence (FA) signals following IC stimulation (*Ibrahim et al., 2017*; *Shibuki et al., 2003*; *Shibuki et al., 2009*; *Husson et al., 2007*) were measured using a stable DC fluorescence illuminator (Prior Lumen 200) and a U-M49002Xl E-GFP Olympus filter cube set (excitation: 470–490 nm, dichroic 505 nm, emission 515 nm long pass, 100 ms exposure time for FA and 5 ms for calcium signals). All data were collected using Retiga EXi camera at a frame rate of 4 Hz for FA and 10 Hz. The time trace of the FA or calcium population signals were obtained by placing regions of interest (ROI) over brain regions (IC, MGB, TRN or AC). The collected time traces were used to compute the Δf/f. For the calcium signals obtained from cortical or thalamic cells, the ROIs were manually made around the cell body. A strong correlation between voltage and calcium signals validated the calcium signals obtained from thalamic cells containing CAL-520 dye after simultaneous whole-cell recording and calcium imaging following the injection of a positive current to the cell (*Figure 5—figure supplement 1*). The average background value was calculated by drawing an ellipse with radii 2.25 times that of the original ROI and subtracting all cell ROI from that ellipse to eliminate overlap. Finally, the Δf/f was computed after 40% of the background value was subtracted from the cell's average value for neuropil correction. Δf/f was computed using the average signal of a 2 s period before the stimulus onset. As illustrated in *Figure 2—figure supplement 2*, the ON-cortical responses were determined following two criteria. Each signal required a rising and falling phase as well as a z-score greater than 3.

## Pharmacological intervention

To inhibit GABAA receptors globally, GABA$_A$-receptor antagonist (*Yaron-Jakoubovitch et al., 2013*), gabazine (Cat# 1262, Tocris) was added to the bath ACSF solution 200 nM, which is near the synaptic IC50 for this compound (*Lindquist et al., 2005*). To specifically inhibit the GABA$_A$ receptors in either MGB or AC, a continuous flow of gabazine (400 nM) was injected into MGB or AC in a counterbalanced manner through a glass pipette (broken tip, 35 µm) which was connected to a picospritzer (Toohey Company, New Jersey, USA). The pipette was filled by a solution composed of 1 ml ACSF +10 µL Alexa Fluor 594 hydrazide, sodium salt dye (Cat#A10438, Invitrogen) to visualize the flow of the solution and to ensure that it was restricted to the site of injection (*Figure 4—figure supplement 1*). The injection was done under 10 psi pressure for 5 min and continuously during imaging. As reported before, to block TRN activity (*Sun et al., 2013*; *Lee et al., 2010a*), the AMPA receptor blocker NBQX (20 µM, Cat# 0373, Tocris) was injected into TRN of the aCTC slice following

the same described procedures. The chemical inhibition of corticothalamic layer 6 cells was conducted by bath perfusion of clozapine-n-oxide (CNO, 5 µM, Cat# 4936, Tocris), the chemical actuator of the chemogenetic probe, hM4Di (*Roth, 2016*) that was solely expressed in corticothalamic layer 6 of *NTSR1*-Cre mouse after viral injection. CNO was also perfused to the imaging chamber containing a slice with a layer 6 expressing m-cherry only as a control for DREADDs.

## Electrophysiology and photoinhibition

Whole-cell recording of cortical layer 4, TRN, or MGB cells was performed using a visualized slice setup outfitted with infrared-differential interference contrast optics. Recording pipettes were pulled from borosilicate glass capillary tubes and had tip resistances of 2–5 MΩ when filled with potassium gluconate based intracellular solution (in mM: 117 K-gluconate, 13 KCl, 1.0 $MgCl_2$, 0.07 $CaCl_2$, 0.1 ethyleneglycol-bis(2-aminoethylether)- N,N,N′,N′-tetra acetic acid, 10.0 4-(2-hydroxyethyl)−1- piperazineethanesulfonic acid, 2.0 Na-ATP, 0.4 Na-GTP, and 0.5% biocytin, pH 7.3, 290 mOsm) for current-clamp mode. Voltage was clamped at −60 mV or +10 mV to measure either the excitatory or inhibitory currents, respectively, using cesium-based intracellular solution (in mM: 117.0 CsOH, 117.0 gluconic acid, 11.0 CsCl, 1.0 $MgCl_2$*$6H_2O$, 0.07 $CaCl_2$, 11.0 EGTA, 10.0 HEPES, pH 7.3, 290 mOsm). Local field potential (LFP) recordings were performed using glass pipette with a broken tip 5–10 µm. LFP signals were filtered offline using Clampfit 10.7 software under Gaussian low pass frequency at 300 Hz as well as a notch filter at 60 Hz. A Multiclamp 700B amplifier and pClamp software (Molecular Devices) were used for data acquisition (20 kHz sampling). To analyze the distribution of the membrane potential of layer 4 cells during the UP state, a time period starting from the stimulation offset and including a similar time of both the cortical upstate event and the basal membrane potential was used. The sampling rate was reduced from 20 kHz to 2 kHz, and spikes and stimulus artifacts were excluded from the analysis.

For photoinhibition, the halorhodopsin eNpHR3.0 probe expressed selectively in either corticothalamic layer 6, layer 5, or IC-GABAergic cells, was activated by illuminating a yellow light (565 nm) obtained from DC fluorescence illuminator (Prior Lumen 200) and Olympus filter cube (U-MF2, Olympus, Japan). The same light was used to illuminate the brain slice with a layer six expressing eYFP only as a control for halorhodopsin. The light was set to illuminate the whole field of the LFP recording chamber using a 4X objective for three seconds extending from one second pre-stimulus and two seconds after the onset of IC stimulation. Based on the initial results related to the peri-stimulus dynamics of corticothalamic layer 6 cells and the post-stimulus cortical activity, 3 s of illumination was chosen to cover the time period one second before the onset of the stimulus as well as the post-stimulus period.

## Histology and confocal imaging

After collecting the aCTC slice from *RBP4*-Cre mouse injected with AAV9-FLEX-EGFP into the AC, the aCTC slice was placed in a 4% paraformaldehyde (PFA) solution for fixation. After one day, the aCTC slice was removed from the PFA and moved to graded sucrose solution (10, 20, and 30%). The aCTC slice was sectioned using a Leica cryostat as 50 µm sections. The sections were imaged using Leica SP8 confocal microscope (excitation: 488 nm and emissions: 515–550 nm).

## Imaging analysis and statistics

Using customized MATLAB codes, all the pseudocolor images (jet colormap) were produced showing the tonotopic map of AC in vivo and the activated brain regions in the aCTC slice. Origin-Pro 2017 software was used to run the statistical tests and generate the graphs. The normality of the distributions of the data was examined using a Kolmogorov-Smirnov test. Accordingly, the suitable parametric (paired t-test or RM-One Way ANOVA test followed by Bonferroni post hoc test) or non-parametric tests (Paired Wilcoxon signed-rank test or Kruskal-Wallis test followed by Dunn's Test) were used. Differences were deemed significant when p-value<0.05. Power analyses were run in G*Power (http://www.gpower.hhu.de). Using an effect size of 50% and alpha <0.05, sample sizes in all experiments provided power to detect significant differences of at least 80%. The animals were randomly allocated in each experimental group.

## Work art

All figures were designed and made using Adobe Illustrator (Adobe, San Jose, CA). To keep working within Adobe environment to avoid losing the resolution of the figures, Adobe Photoshop (Adobe, San Jose, CA) was used to crop the borders of some images to save space, draw scale bars, and adjust brightness and gamma balance of grayscale images showing the electrophysiological traces. All manipulations in brightness/contrast/gamma were uniform across the entire image.

## Acknowledgements

We thank Dr. Murray Sherman (The Department of Neurobiology, University of Chicago), Dr. Shane Crandall (The Department of Physiology, Michigan State University), and Dr. Brian Theyel (The Department of Neuroscience, Brown University) for their valuable comments on the manuscript.

## Additional information

### Funding

| Funder | Grant reference number | Author |
| --- | --- | --- |
| NIDCD | R01DC013073 | Daniel A Llano |
| NIDCD | R21DC014765 | Daniel A Llano |
| National Science Foundation | 1515587 | Daniel A Llano |

The funders had no role in study design, data collection and interpretation, or the decision to submit the work for publication.

### Author contributions

Baher A Ibrahim, Conceptualization, Data curation, Formal analysis, Investigation, Visualization, Methodology, Writing - original draft, Writing - review and editing, Performing most of the experiments; Caitlin A Murphy, Validation, Investigation, Methodology, Performing some experiments; Georgiy Yudintsev, Investigation, Methodology; Yoshitaka Shinagawa, Software, Formal analysis, Writing - review and editing; Matthew I Banks, Validation, Investigation, Methodology; Daniel A Llano, Conceptualization, Resources, Data curation, Formal analysis, Supervision, Funding acquisition, Validation, Investigation, Visualization, Methodology, Project administration, Writing - review and editing

### Author ORCIDs

Baher A Ibrahim (iD) https://orcid.org/0000-0002-0062-7589
Caitlin A Murphy (iD) http://orcid.org/0000-0002-6319-9470
Daniel A Llano (iD) https://orcid.org/0000-0003-0933-1837

### Ethics

Animal experimentation: This study was performed in strict accordance with the recommendations in the Guide for the Care and Use of Laboratory Animals of the National Institutes of Health. All of the animals were handled according to approved institutional animal care and use committee (IACUC) protocols (#18236) of the University of Illinois at Urbana-Champaign. All surgery was performed under anesthesia, and every effort was made to minimize suffering.

### Decision letter and Author response

Decision letter https://doi.org/10.7554/eLife.56645.sa1
Author response https://doi.org/10.7554/eLife.56645.sa2

## Additional files

### Supplementary files
• Transparent reporting form

### Data availability
All data generated or analyzed during this study are included in the manuscript and supporting files. Source data files have been provided for Figures 1, Figure 2, Figure 2-figure supplement 2, Figure 2-figure supplement 6, Figure 3, Figure 4, Figure 5, Figure 6, Figure 6-figure supplement 1. The datasets are available at Dryad under a DOI (https://doi.org/10.5061/dryad.qrfj6q5c4).

The following dataset was generated:

| Author(s) | Year | Dataset title | Dataset URL | Database and Identifier |
|---|---|---|---|---|
| Ibrahim BA, Murphy CA, Yudintsev G, Shinagawa Y, Banks MI, Llano DA | 2020 | Corticothalamic gating of population auditory thalamocortical transmission in mouse | https://doi.org/10.5061/dryad.qrfj6q5c4 | Dryad Digital Repository, 10.5061/dryad.qrfj6q5c4 |

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
