## [Decision Letter]

**Acceptance summary:**

This study employs an impressive combination of in vivo and in vitro methods to investigate the neural circuitry that determines whether neurons in auditory cortex respond or not to relatively quiet sounds. The results suggest that layer 6 corticothalamic feedback via the thalamic reticular nucleus is responsible for gating cortical population responses, proving new insight into the role of descending projections from the auditory cortex to the thalamus.

**Decision letter after peer review:**

Thank you for submitting your article "Corticothalamic gating of population auditory thalamocortical transmission in mouse" for consideration by *eLife*. Your article has been reviewed by 3 peer reviewers, and the evaluation has been overseen by Andrew King as the Senior Editor and Reviewing Editor. The following individual involved in review of your submission has agreed to reveal their identity: Daniel B Polley (Reviewer #2).

The reviewers have discussed the reviews with one another and the Reviewing Editor has drafted this decision to help you prepare a revised submission.

Summary:

This is an impressive paper, which utilizes an array of state-of-the-art tools to dissect the contribution of cortical feedback to the thalamic reticular nucleus in the omission of responses to sound in the auditory cortex. This is an important and surprising finding, which provides new insight into the role of corticothalamic circuits. These findings are based on a combination of in vivo and in vitro population recordings and make excellent use of a powerful slice preparation that allows the connectivity of the midbrain, thalamus and cortex to be explored. Although the reviewers were enthusiastic about much of the study, they thought that the findings were not well presented and disagreed with some of your interpretations of the data. In particular, they were unanimous in their view that the sections dealing with gamma oscillations, thalamic synchrony and deep learning are weak and incomplete, and should be removed from the paper. They were also concerned that critical control groups (the use of a control fluorophore in the case of the halorhodopsin experiments and a CNO only group for the DREADDS study) are not reported. This is particularly important where such large injection volumes were used in very young mice. Consequently, it was felt that additional data as well as re-analysis and re-writing would be needed before this paper could be considered acceptable for publication.

*Reviewer #1:*

This is a nice paper that proposes a mechanism that allows for gating of population cortical activity. The paper uses a combination of in vivo and in vitro population recordings that are backed up with some excellent slice physiology from a courageous colliculo-thalamocortical slice preparation.

Line 69. Why must these hierarchical filters be linear?

Line 73. This formed complex hallucinations argument isn't really clear. Are you just arguing that a hierarchical linear filter model cannot explain anything that causes elevated activity in primary sensory cortices, that isn't present in subcortical stations?

Line 85. I'm not sure I agree with the statement that the thalamus is strikingly similar across modalities. The two references that are cited only discuss canonical cortical circuits.

Line 91. The statement "all-or-none" to describe the population response seems inadequate, given the variability in "ON" responses in Figure 1D.

Figure 1. Where do these ON/OFF responses fall within the trial presentations – are there any time dependence that could reflect photobleaching? There is also a large fraction of OFF responses that seem to be driven below baseline (in Figure 1D) – is this an artifact of the dF/F computation? On that note, the methods do not explain how you computed dF/F (there are different ways that one can do this).

Line 105. 37dB is quite a specific intensity – why did you choose this?

Line 108 and Line 110. You say "to 10 presentations of the same tone", and "across the 40 trials of the same sound presentation". It's not quite clear what is meant.

Line 123. Activating the IC and observing AC response is due to di-synaptic circuitry. Therefore, does the AC "failure rate" change with the intensity of the stimulation? Can you stimulate hard enough to overcome the thalamic inhibition that you claim mediates the OFF responses?

Figure 2E. Why is the dF/F in the MGB orders of magnitude lower than IC or ACtx?

Line 139. It's not immediately clear to me how are you getting single-cell resolution using this one-photon preparation? The images shown throughout are "wide field" population images. Are you able to show any cellular examples? Are you using some kind of NMF algorithm to achieve cellular resolution? How are you controlling for issues like neuropil contamination (that are ubiquitous in 2P data).

Line 146. On the issue of up states vs down states. If you histogram the membrane potential, is it bimodal to reflect the different states?

Line 155-160. This isn't particularly well described. What exactly is it being modeled here? It seems like you are performing a binary classification, e.g. using either MGB activity or striatum activity to predict whether the corresponding cortical response was ON or OFF? Why does this imply that network-level activity may determine whether an on or off response occurred? Why is the striatum region irrelevant (i.e. it is still (indirectly) connected the cortex via the thalamus, no?). Why the need to use such an advanced machine learning method? Did simpler binary classification tools fail? Does this method also give the same result when used on the GCaMP or whole cell recordings?

Figure 3B. Why are there are more OFF responses that appear in the wash condition? Can you add a third variable to Figure 3C to show this – shouldn't the wash be statistically indistinguishable from control?

Line 195. This is actually a question regarding choices throughout the paper – what dictated the choice of either flavoprotein or calcium? The deep learning results referenced here were established with flavoprotein, but these follow-up experiments use calcium.

Figure 4B. It's not quite clear what is going in this figure. Would some kind of population signal help? Or an alternative way to plot the data? Do these individual traces all have well-defined peak latencies (it's hard to tell from the figure). You're showing the variance of the peak latencies – but couldn't you just compare the peak latencies directly? Does it give the same result? Also, you talk about three categories in the text, but are only showing two categories in the figure. A latency analysis is also a little bit difficult to interpret with calcium data (both from the calcium timescale and the acquisition frame rate) – it might be a good idea to either discuss these limitations, since you are arguing for synchronicity based on a small reduction in variance within this data.

Line 195. "Calcium signals of MGB cells were imaged following the stimulation of the IC during ON vs OFF cortical responses" – is it not the IC stimulation that is causing the ON or OFF cortical responses?

Line 373 – "The detailed procedures have been described before (77)". This seems like it may be the wrong citation? (77) is a theoretical conference paper with no in vivo imaging details.

Line 220 – Is there a particular reason to use neonates for the experiments in this section?

Figure S7. I'm slightly confused by the MGB of the RBP4-Cre mouse – it looks like there is no TRN labelling but also no MGB labelling?

Line 270. It's not possible to understand what you did here without jumping to the methods.

Line 274. Is there an intuitive reason why a shorter pre-stimulus period (which should contain more oscillatory information) be less predictive?

Line 276. "A classifier?" Can you justify your classifier choice here, compared to the more complicated deep learning approach used for classification earlier in the paper?

Figure S8. Based on the text, the gamma prediction is important to the circuit story. It may be a better idea to include it as a main figure?

*Reviewer #2:*

There is much to admire in the paper by Ibrahim and colleagues. They are the only lab in the world (to the best of my knowledge) that has worked out the geometry of a brain slab that leaves the synaptic connections intact from the midbrain-thalamus-MGB-TRN-cortex-back to TRN/MGB. Here, they use this slab to great effect by revitalizing the old cortical up/down state phenomenon in the context of all-or-nothing responses in A1 to stimulation of midbrain afferents. They have convinced me that the basic phenomenon is real and that it is interesting. Further, I am convinced that they can reduce the probability of OFF response trials by: (1) blocking GABA_A_ receptors, (2) blocking AMPA receptors in TRN, (3) inactivating Ntsr1-Cre L6 CT neurons. They also have data to show that (4) inactivating feedforward inhibition from the IC or glutamatergic L5 projection neurons does not affect OFF response probability.

While there is a lot of good in this manuscript, there is plenty of bad and ugly as well. In their efforts to weave a complete theory of the phenomenon, they are forced to rely on a few very weak/poorly developed pieces of data. Emphasizing these more tenuous findings (Figure 4, Figure S4, Figure S8), ends up doing them a disservice because it takes away from their more solid observations. Apart from eliminating the dubious findings, the figures and the analysis are poorly organized, confusing and do not convey the process of processing the data from example cases to summary plots that relate to their statistics. They look more like lab meeting figures than manuscript figures.

In the final analysis, I think the work offers a significant conceptual advance based on an innovative technical approach. The manuscript can be substantially improved through eliminating the under-developed analyses, reorganizing the remaining figures and adding some additional analysis. The authors needn't feel compelled to oversell the reader on a complete comprehensive story when some of holes are not yet filled in. It's fine that some of the mechanisms are unknown so long as the shortcomings and caveats are clearly identified.

1. Figure 1 is potentially important but is not too convincing in its current form. One reason that it's important is that the rest of the manuscript is a developmental neuroscience study that is never acknowledges this fact. The intro and discussion with the talk of schizophrenia and hallucinations are too far-ranging for my taste, but particularly in the context that their study was performed in mice that had only been connected to the auditory world for less than 1 week. It would be very easy to write off the phenomenon they describe here as an oddity o development that would not apply to the mature cortex.

For me, this underscores the potential importance of Figure 1, where they demonstrate the phenomenon in vivo using older animals. The biggest impediments in its current form are (i) the age of the animals used for transcranial imaging is never mentioned. I presume they are adult? (ii) The entire analysis is based on a single tone frequency presented a single sound intensity that is very near threshold (5k at 37 dB SPL). This makes me question the generality of the principle described here. Having done this trial-by-trial analysis myself on this type of data, I don't think it really works for strong, suprathreshold stimulus intensities. Is this entire phenomenon restricted to a very narrow range of near-threshold intensities? In that case…is it really a phenomenon or just what happens near minimally effective stimulation levels (which is already very well known). Obviously, the same critique applies to the current levels they use for the rest of the study. It would have been nice to see a more systematic and quantitative examination of the stimulus parameters where this phenomenon is strong versus weak.

2. Imaging from "single cells". Single cell imaging is not possible with epifluorescence illumination in a slice this thick, especially without neuropil removal. At best they are ROIs but even then, what is the value when there is no basis for the argument that their ROIs are independent response units? I suggest eliminating 2G, Figure 4 and Figure S4. The issues with Figure 4 (and S4) go way beyond the cellular resolution issues, they are uninterpretable for a number of reasons.

3. In Figure 2D, I'm not convinced that they are looking at signals from TRN cell bodies as opposed to thalamocortical/corticothalamic axons. The thalamic radiation forms a dense fascicle right at the point of the TRN in this plane so it could be very hard to distinguish between TRN cell bodies and axon signals with the optics they are using. Too bad they didn't express GCaMP in the GAD-Cre mice they use for inhibiting the feedforward inhibitory IC projection neurons.

4. Calcium imaging in the MGB and IC. Looking at other publications of the Thy1-GCaMP6 mouse, I see lots of labeling in pyramidal neurons in neocortex and hippocampus, but no labeling in the MGB or IC. I'm confused about what is being measured in Figure 2E. Maybe there is expression in younger mice or is just very faint?

5. Figure 3E-H (graphics could be greatly improved here) show that EPSCs in MGB are unchanged in cortical ON vs OFF trials but IPSCs are elevated in OFF trials. This is one of the mysteries of their study that they try to address with a bunch of uninterpretable analysis methods (Figure 4 and Figure S4). The conundrum they face is how does enhanced thalamic inhibition create a cortical OFF response without making the MGB excitatory response weaker? Figure 4 and S4 attempt to explain this but are contrived and unconvincing. I don't disagree but I would suggest that they are thinking about it too simplistically. It doesn't matter what the MGB cell bodies are doing, what matters is what they are transmitting to the cortex during OFF trials. It would be helpful if they quantified these data more carefully by looking at the relative timing of the IPSC and EPSC responses. Looking at the study by Reinhold, Lien and Scanziani 2015, the TRN can regulate sustained spiking and thalamocortical synaptic depression in ways that cannot be dismissed just by measuring the excitatory response. They don't have to fully answer this here. They can do the best with whatever straightforward analysis they can manage and then devote some space in the discussion to actually discussing alternative explanations for their experiments instead of rhapsodizing about how these data solve the greatest mysteries of the brain.

6. Figure 3K – It would be important to see the individual data points. The legend says the data are an n=5, reflecting 5 slices from 4 animals. How do you get two of these slices from one animal? Regardless, this is a very small sample to make such a strong conclusion as blocking cortical GABA_A_ receptors does not affect the OFF response probability. Was the AC Gabazine always performed after the MGB wash? The order should be counterbalanced and really I'm not even sure what the justification is for using a t-test on a sample this small. Unclear how the assumptions of the test can really be determined.

7. I suggest putting all three of their halorhodopsin experiments into one figure (IC inhibitory feedforward neurons, L6 CT and L5 projection neurons). I'm also bothered that there isn't a control condition where mice have undergone a perinatal injection to express a control fluorophore. This seems particularly warranted because they injected an enormous volume of virus solution (1600 nL) into very small area. Apart from this control, while I think the L6 inactivation result is very interesting, I strongly disagree with their often repeated interpretation of the data, that "Population OFF responses were associated with TRN-mediated inhibition at the level of the MGB, under the control of layer 6 corticothalamic projections".

Their data do not support this conclusion. Yes, blocking AMPA receptors in TRN reduced the probability and Yes, silencing L6 CT also reduced the probability but it is a logical fallacy to assume those are causally related. Ntsr1 neurons make synaptic contacts on TRN neurons, MGB neurons, local AC inhibitory neurons and local AC excitatory neurons. It is entirely possible that the Ntsr1 neurons are mediating this effect through local connections in the cortex and not through the descending projection. In that regard, it is a bit frustrating that the main point of the Guo 2017 study was the Ntsr1 neurons can strongly bias AC neurons to become less responsive to afferent activity via intracortical connections, yet these findings are not mentioned in this context.

Don't get me wrong, they might be right. But providing that the L6 CTs modify MGB via TRN would require a different type of experiment to isolate the influence of L6 CT axons onto TRN neurons and not the other three types of neurons that it communicates with. Again, good topic for a revamped Discussion section.

8. Figure S3 should be upgraded to a main figure and quantified.

9. Figure S6 is interesting but preliminary. There are only four data points and DREADD experiments are usually considered uninterpretable without a control condition in which CNO is applied with the expression of a control fluorophore (i.e., without the designer receptor) to account for known off-target effects of CNO.

10. The issues with Figure S7 are that Rbp4-Cre does not selectively label "PT-like" L5 cells that project sub-cerebrally. In fact, most of their axons remain within the cortex. To this point, the expression they show in no way resembles the cartoon in Panel A but more problematically does not compare to the expression in the Ntsr1 neurons. As such, it is hard to interpret much from the negative result because it may be that they are hyperpolarizing many fewer neurons overall and very often not the L5 neurons that project to the MGB.

11. Figure S8 is another example of them trying to squeeze too much out of their data. First of all, I was unclear about how they were confident that they patched Ntsr1 neurons and not other L6 cell types. But mainly the analysis methods were a little strange to me. Why not just filter in the Gamma band (or other bands)? Also there is a huge literature showing the phase of ongoing 2-6Hz oscillations gate the response probability of AC neurons (e.g., in Guo 2017). It would have been reasonable calculate the phase and amplitude of the pre-stimulus membrane currents from its analytical signal using the Hilbert transform. In its current form, the quantification in this figure is not convincing at all.

Apart from that, the Pre-stimulus L6 neuronal activity could only accurately predict 58.1 {plus minus} 3.5 SD% ON and OFF cortical responses, which is subtle compared to the theoretical random accuracy 55.1%. Moreover, should the random accuracy be calculated by training and testing the model with shuffled ON and OFF responses?

*Reviewer #3:*

This is an interesting paper that finds that the cortical feedback to the thalamic reticular contributes to the omission of responses in the cortex. This is an important and surprising finding, and the authors bring a whole arsenal of state-of-the-art tools to dissect the contribution of this pathway. The variety of the methods, that include calcium imaging, flavoprotein imaging, electrical stimulation, whole cell recording, pharmacologic/optogenetic/chemogenetic manipulations, all point towards the same conclusions. The data that are presented support most of the claims of the paper. Most of my concerns reflect the interpretation of the data, the focus of the introduction and discussion, some missing controls and the lack of report of statistics in the main body. I believe that the authors should be able to address these concerns with the data in hand, through additional analysis and re-interpretation.

1. The emphasis in the introduction and discussion on the "classical models" of purely feedforward hierarchical processing seems somewhat forced. Numerous studies, especially in the auditory system, have addressed the importance of feedback in anatomy, auditory processing, learning and behavior. Rather than skipping over this important work, the paper would benefit from the discussion of previous work on the function of feedback in sensory processing. This would furthermore allow the authors to better define what role their work plays in this context.

2. Similarly, the emphasis on the role of this feedback in neuronal oscillations seems to be overstated. In the introduction, the discussion of hallucinations seems to come out of nowhere. In the paper, only one supplementary figure is devoted to analysis of oscillations, whereas in the abstract, introduction and discussion, these is an extensive focus on the gamma oscillations. The gamma activity -based prediction of ON vs OFF responses seems to be weak. The classifier only had an accuracy of 58%, which is not very high, and is only barely above random accuracy of 55. This focus and confusing result detracts from the main message of the paper.

3. The terminology of ON and OFF responses seems to be at odds with the accepted terminology in the auditory field. Typically, in the auditory field, ON responses refer to the activity at the onset of a stimulus, and OFF responses refer to the activity after stimulus offset. OFF responses, where there is no response at all, is confusing. The authors allude to the use of UP and DOWN states in the literature, but there is only indirect evidence that this activity indeed corresponds to those states. Perhaps using "Response" and "Omission"; yes/no; 1/0 or +/- could work.

4. The claim that the thalamus selects co-activated cortical ensembles, referred to in the abstract, last sentence of the introduction as well as the discussion, is not supported by data. Perhaps it is known that the auditory cortex uses partially overlapping ensembles of coactivated neurons to represent auditory stimuli, but that data is not represented in this paper. In fact, the data in this paper hinges on an "all or nothing response," which undermines this idea of coactivated ensembles in the AC. These contrasting ideas deserve reconciliation.

5. Statistics for all claims should be reported in the main text. Some statistics are missing, for example, in Figure 5D, the distribution of the histograms.

6. It is great that the paper starts with results in vivo. However, those are under anesthesia. Are OFF responses also present in awake animals, perhaps influenced by attention or arousal levels? Perhaps worth speculating in the discussion on the functional importance of this corticothalamic gating in awake behaving animals.

7. The authors use a deep learning algorithm to establish the predictive power of the response, training on MGB data vs irrelevant striatum. How would the algorithm work for IC data? Furthermore, it is unclear whether deep learning is required for this analysis. Would more simple statistical measures, such as correlation or Granger causality measures work here as well? Can the predictive ability of layer 6 activity be improved with just using pre-stimulus layer 6 firing rate, as opposed to LFP power spectra?

8. The authors observed cortical IPSCs during ON responses but not OFF responses. The importance of this finding would be better demonstrated by comparing it with cortical EPSCs. Presumably, ON responses are characterized by large EPSPs and medium IPSCs, whereas OFF responses are characterized by no EPSCs or IPSCs. That way, it's clearer that OFF responses are not simply caused by increase in IPSC magnitude. It could then be contrasted that with the increase in IPSC magnitude observed in MGB, as they show in figure 3G. The importance of figure 3E/3F is unclear. It would be great to explain why IPSCs are observed in cortical ON responses and not OFF responses.

9. How well do gabazine and NBQX stay localized to the area of injection? Is there any chance of diffusion of the agent confounding results? How was localization confirmed?

10. What are the controls for chemogenetics?

11. In Figure 4, when performing calcium imaging in MGB, which trials had cortical ON vs OFF responses? Was that determined through simultaneous LFP recording in AC (diagram in figure 4A)?

12. How well can latency of response and response timing variability (Figure 4) be assessed using GCaMP6s, which has a response time scale on the order of seconds? Is the variability in response timing really on the order of hundreds of milliseconds (Figure 4D)? Perhaps this would be better evaluated using whole cell recording or electrophysiology.

13. Is it MGB synchrony, or MGB inhibition, that leads to cortical ON vs OFF responses? The importance of MGB synchrony, or at least its interaction with TRN-MGB inhibition, is unclear. Furthermore, the synchrony analysis should be conducted with more appropriate statistical tools, including correlation measures. It remains unclear how a desynchronized MGB leads to an OFF response (aka no firing) in the cortex. Do layer 4 cortical cells not reach firing threshold when the inputs are asynchronous? Or do cortical neurons not receive inputs from MGB when the neurons fire asynchronously (due to further gating by third party cells)?

14. Is the difference in power at ~36Hz significant when controlling for multiple comparisons across 11 different LFP power spectra frequencies? Perhaps the authors could try running a 2-way ANOVA with frequency and ON/OFF status as factors.

15. I recommend tightening the writing style throughout: e.g. in abstract: "Here, we elucidate the mechanism for gating of population activity." Some paragraphs are multiple pages long. There are run-on sentences, and the authors use passive tense extensively. I would recommend splitting up the paragraphs, so that each paragraph contains one result/analysis, as well as two sentences for motivation and conclusion.

[Editors' note: further revisions were suggested prior to acceptance, as described below.]

Thank you for resubmitting your work entitled "Corticothalamic gating of population auditory thalamocortical transmission in mouse" for further consideration by *eLife*. Your revised article has been evaluated by Andrew King as the Senior Editor and Reviewing Editor.

The authors are to be commended for carrying out a number of new experiments to provide essential control data and to provide additional information that has strengthened the study. The majority of the other concerns raised by the reviewers have been dealt with adequately, but there are a few remaining issues that need to be addressed.

Both the in vivo and slice data now clearly illustrate the dependence of the cortical responses on the stimulus magnitude, with the probability of an OFF response occurring declining with increasing stimulus strength. This reinforces one of the points made by reviewer 2, namely that these effects appear be present only for near minimally effective stimulation levels. That raises the question of what function they serve and how these responses contribute to our understanding of cortical processing under more natural conditions (including in awake animals at higher sound levels). These are important issues that need to be addressed in the Discussion.

The readability of the text would benefit further from greater use of paragraphs (as pointed out by reviewer 3). For example, one paragraph runs from lines 304-380. This does include some deleted text, but is still far too long.

Line 364: 5J should be 6J.

Line 423: "exposure to pure tones" might be construed as a form of passive acoustic environment. "Presentation of pure tones" would be less ambiguous.

Figure 1 legend: there are several full stops missing.

Figure 2 – Supplementary Figure 3: the legend refers to red and green boxes, but there are none in the figure (at least in the composite pdf of the manuscript).

Line 943: the sentence ends "showing".

Figure 6 – Supplementary Figure 1G: all traces are black, whereas the legend claims that some are brown or green.

---

## [Author Response]

Reviewer #1:This is a nice paper that proposes a mechanism that allows for gating of population cortical activity. The paper uses a combination of in vivo and in vitro population recordings that are backed up with some excellent slice physiology from a courageous colliculo-thalamocortical slice preparation.Line 69. Why must these hierarchical filters be linear?

We apologize for the lack of clarity. Indeed, we do not believe that hierarchical filters need to be linear. However, we argue that classical models of sensory processing have relied on combinations of linear filters that are arranged hierarchically. In the current case, we in fact find a strongly nonlinear response in the cortex in response to midbrain stimulation (revised Figure 3). We have added the qualifier “classical” to our statement about traditional views of sensory processing.

Line 73. This formed complex hallucinations argument isn't really clear. Are you just arguing that a hierarchical linear filter model cannot explain anything that causes elevated activity in primary sensory cortices, that isn't present in subcortical stations?

We argue that classical models of sensory processing cannot explain the presence of formed patterns of cortical activity in the absence of subcortical activity. In retrospect, although our current data may lead to a greater understanding of how patterns of responses occur in the auditory cortex, our data do not address how hallucinations are formed. What we speculate here is that the bottom-up flow of sensory information could activate stereotyped cortical populations of neurons. We have thus toned down our discussion of the potential implications of the current work with respect to hallucinations.

Line 85. I'm not sure I agree with the statement that the thalamus is strikingly similar across modalities. The two references that are cited only discuss canonical cortical circuits.

We have changed “strikingly similar” to “relatively homogeneous” and have added another more recent reference to support this statement.

Line 91. The statement "all-or-none" to describe the population response seems inadequate, given the variability in "ON" responses in Figure 1D.

Although the ON cortical events showed some variability, the responses fell into two discrete categories. We have added the text “, with some variability within each class (Figure 1C)” to indicate that there is variability within the classes.

Figure 1. Where do these ON/OFF responses fall within the trial presentations – are there any time dependence that could reflect photobleaching? There is also a large fraction of OFF responses that seem to be driven below baseline (in Figure 1D) – is this an artifact of the dF/F computation? On that note, the methods do not explain how you computed dF/F (there are different ways that one can do this).

This is an excellent question. We note that the blue excitation light was not on during the whole presentation sequence. It was turned off between presentations and turned on 5 seconds prior to the stimulus onset, precisely to avoid bleaching. In addition, we interleave stimulus trials with blank trials and subtract the blank traces to eliminate any residual photobleaching effects, which can also produce negative values for dF/F. We have now added a figure to show that there are no apparent order effects in our data and added text to clarify the method for computing ∆F/F.

Line 105. 37dB is quite a specific intensity – why did you choose this?

We have found that the population ON/OFF phenomenon is seen near response threshold. Below threshold, no responses are found and above threshold, consistent full responses are found in the AC. In the case of 37 dB SPL, we did use higher intensities and found consistent ON responses. We have modified Figure 1 and the text to indicate that the presence of ON and OFF responses appears to be restricted to near-threshold stimuli.

Line 108 and Line 110. You say "to 10 presentations of the same tone", and "across the 40 trials of the same sound presentation". It's not quite clear what is meant.

We apologize for this error. We modified the figure to have an average image for the 40 trials.

Line 123. Activating the IC and observing AC response is due to di-synaptic circuitry. Therefore, does the AC "failure rate" change with the intensity of the stimulation? Can you stimulate hard enough to overcome the thalamic inhibition that you claim mediates the OFF responses?

Thank you for raising this issue. As stated above, the presence of ON/OFF responses does indeed depend on stimulus amplitude. The ON/OFF phenomenon is seen just above threshold and at high stimulus amplitudes, only ON responses are seen. We have added Figure 3 and corresponding text to address this issue.

Figure 2E. Why is the dF/F in the MGB orders of magnitude lower than IC or ACtx?

Excellent observation. We have previously shown that flavoprotein autofluorescence responses in the thalamus are significantly smaller than corresponding responses in the cortex (Llano et al. 2009, PMID: 19321634). We do not know the reason for the smaller responses, but speculate that because the flavoprotein signal is derived primarily from neuropil (where the majority of signal-generating mitochondria are), that brain regions with strongly aligned neuropil processes (like apical dendrites in pyramidal cells) produce stronger signals. We do note that the flavoprotein responses in the MGB, though small, are consistent and align well with the electrophysiological results presented in Figure 4. We have added clarifying text to the revised manuscript.

Line 139. It's not immediately clear to me how are you getting single-cell resolution using this one-photon preparation? The images shown throughout are "wide field" population images. Are you able to show any cellular examples? Are you using some kind of NMF algorithm to achieve cellular resolution? How are you controlling for issues like neuropil contamination (that are ubiquitous in 2P data).

Respectfully, we submit that single-cell analysis of 1-photon calcium imaging has been routinely done for some time (Maclean et al. 2005 PMID: 16337918, Berger et al. 2007 PMID: 17360827, Cameron et al. 2016 PMID: 27183102). We have now re-analyzed the data using neuropil correction and find similar results, and present modified data in Figures 2G and 5.

Line 146. On the issue of up states vs down states. If you histogram the membrane potential, is it bimodal to reflect the different states?

Thank you for the suggestion. We have made histograms of the membrane potential of layer 4 cells, and it clearly showed a bimodal distribution that implicates the different states of the cellular activity. The histograms are now in a supplementary figure (Figure 2—figure supplement 6).

Line 155-160. This isn't particularly well described. What exactly is it being modeled here? It seems like you are performing a binary classification, e.g. using either MGB activity or striatum activity to predict whether the corresponding cortical response was ON or OFF? Why does this imply that network-level activity may determine whether an on or off response occurred? Why is the striatum region irrelevant (i.e. it is still (indirectly) connected the cortex via the thalamus, no?). Why the need to use such an advanced machine learning method? Did simpler binary classification tools fail? Does this method also give the same result when used on the GCaMP or whole cell recordings?

Complying with the other reviewers’ and editor’s suggestions, we have removed this section from the manuscript.

Figure 3B. Why are there are more OFF responses that appear in the wash condition? Can you add a third variable to Figure 3C to show this – shouldn't the wash be statistically indistinguishable from control?

We apologize for the confusion. We have now compared the wash vs. baseline responses, and the differences are not significant. In revised figure 3C (now Figure 4C), we now use one-way ANOVA instead of paired t-test, and the test showed a nonsignificant difference between the control and wash groups.

Line 195. This is actually a question regarding choices throughout the paper – what dictated the choice of either flavoprotein or calcium? The deep learning results referenced here were established with flavoprotein, but these follow-up experiments use calcium.

We apologize for the confusion. The main imaging technique used in the paper is flavoprotein autofluorescence imaging because it does not require any modifications to the slice or to the animal. Calcium imaging was used to demonstrate that the ON/OFF phenomenon was not an artifact of the use of a metabolic signal, as well as to examine the cellular activity in layer 4 cells and MGB cells. The deep learning component of the manuscript has been removed, as suggested by the other reviewers and editor. We have revised the text accordingly.

Figure 4B. It's not quite clear what is going in this figure. Would some kind of population signal help? Or an alternative way to plot the data? Do these individual traces all have well-defined peak latencies (it's hard to tell from the figure). You're showing the variance of the peak latencies – but couldn't you just compare the peak latencies directly? Does it give the same result? Also, you talk about three categories in the text, but are only showing two categories in the figure. A latency analysis is also a little bit difficult to interpret with calcium data (both from the calcium timescale and the acquisition frame rate) – it might be a good idea to either discuss these limitations, since you are arguing for synchronicity based on a small reduction in variance within this data.

We apologize for the lack of clarity. Figure 4B was meant to show an overlay of all of the calcium traces during ON vs. OFF response. Figures 4B-4D are alternative ways of showing the same data. We respectfully would like to keep Figure 4B (now Figure 5E) because it shows the reader the raw traces and illustrates the increased synchrony during ON responses. Revised Figure 5A-C is taken from the latency of the responses as suggested. We agree that extracting latency data from calcium traces has significant limitations, which we now discuss in the revised manuscript.

Line 195. "Calcium signals of MGB cells were imaged following the stimulation of the IC during ON vs OFF cortical responses" – is it not the IC stimulation that is causing the ON or OFF cortical responses?

Thank you for pointing this out. The reviewer is correct and we have re-written this sentence for clarity as “To test this hypothesis, the time courses of calcium signals of MGB cells were imaged following the stimulation of the IC, and were compared during population ON vs. population OFF cortical responses.”

Line 373 – "The detailed procedures have been described before (77)". This seems like it may be the wrong citation? (77) is a theoretical conference paper with no in vivo imaging details.

Reference 77 (now reference 71) refers to a book chapter that we have written that describes the details of the imaging procedure.

Line 220 – Is there a particular reason to use neonates for the experiments in this section?

To obtain an aCTC slice with intact connection between IC, MGB, and AC, the slice should be taken from P15-18 mouse pups. As such, the viruses should have been injected into P4 neonates to allow a good time for gene expression the virus carries. We have added clarifying text to the revised manuscript and added more discussion about the issue that the slice work was all done in neonates.

Figure S7. I'm slightly confused by the MGB of the RBP4-Cre mouse – it looks like there is no TRN labelling but also no MGB labelling?

Layer 5-labeled terminals in the thalamus are sparse (Llano and Sherman 2008, PMID: 18181153), so at this magnification one cannot see them. We have added clarifying language, and, based on suggestions from Reviewer 2, an additional control which is to use a virus that does not express halorhodopsin and have added a figure that shows a high-magnification view of thalamic terminals present in the slice after layer 5-RBP4 labeling. The new data are in Figure 6—figure supplement 2.

Line 270. It's not possible to understand what you did here without jumping to the methods.

Based on the suggestion of the other reviewers and editor, this part of the manuscript was removed.

Line 274. Is there an intuitive reason why a shorter pre-stimulus period (which should contain more oscillatory information) be less predictive?

Based on the suggestion made by the other reviewers, this section was removed.

Line 276. "A classifier?" Can you justify your classifier choice here, compared to the more complicated deep learning approach used for classification earlier in the paper?

Based on the suggestion made by the other reviewers, this section was removed.

Figure S8. Based on the text, the gamma prediction is important to the circuit story. It may be a better idea to include it as a main figure?

Based on the suggestion made by the other reviewers, this section was removed.

Reviewer #2:There is much to admire in the paper by Ibrahim and colleagues. They are the only lab in the world (to the best of my knowledge) that has worked out the geometry of a brain slab that leaves the synaptic connections intact from the midbrain-thalamus-MGB-TRN-cortex-back to TRN/MGB. Here, they use this slab to great effect by revitalizing the old cortical up/down state phenomenon in the context of all-or-nothing responses in A1 to stimulation of midbrain afferents. They have convinced me that the basic phenomenon is real and that it is interesting. Further, I am convinced that they can reduce the probability of OFF response trials by: (1) blocking GABA_A_ receptors, (2) blocking AMPA receptors in TRN, (3) inactivating Ntsr1-Cre L6 CT neurons. They also have data to show that (4) inactivating feedforward inhibition from the IC or glutamatergic L5 projection neurons does not affect OFF response probability.While there is a lot of good in this manuscript, there is plenty of bad and ugly as well. In their efforts to weave a complete theory of the phenomenon, they are forced to rely on a few very weak/poorly developed pieces of data. Emphasizing these more tenuous findings (Figure 4, Figure S4, Figure S8), ends up doing them a disservice because it takes away from their more solid observations. Apart from eliminating the dubious findings, the figures and the analysis are poorly organized, confusing and do not convey the process of processing the data from example cases to summary plots that relate to their statistics. They look more like lab meeting figures than manuscript figures.In the final analysis, I think the work offers a significant conceptual advance based on an innovative technical approach. The manuscript can be substantially improved through eliminating the under-developed analyses, reorganizing the remaining figures and adding some additional analysis. The authors needn't feel compelled to oversell the reader on a complete comprehensive story when some of holes are not yet filled in. It's fine that some of the mechanisms are unknown so long as the shortcomings and caveats are clearly identified.1. Figure 1 is potentially important but is not too convincing in its current form. One reason that it's important is that the rest of the manuscript is a developmental neuroscience study that is never acknowledges this fact. The intro and discussion with the talk of schizophrenia and hallucinations are too far-ranging for my taste, but particularly in the context that their study was performed in mice that had only been connected to the auditory world for less than 1 week. It would be very easy to write off the phenomenon they describe here as an oddity o development that would not apply to the mature cortex.For me, this underscores the potential importance of Figure 1, where they demonstrate the phenomenon in vivo using older animals. The biggest impediments in its current form are (i) the age of the animals used for transcranial imaging is never mentioned. I presume they are adult?

Thank you for making these points. Yes, the animal shown in Figure 1 is an adult animal, which has now been described in the revised text. We have also added text to describe the limitations on the interpretation of the study, given the finding that the animals in the slice part of the study were young.

(ii) The entire analysis is based on a single tone frequency presented a single sound intensity that is very near threshold (5k at 37 dB SPL). This makes me question the generality of the principle described here. Having done this trial-by-trial analysis myself on this type of data, I don't think it really works for strong, suprathreshold stimulus intensities. Is this entire phenomenon restricted to a very narrow range of near-threshold intensities? In that case…is it really a phenomenon or just what happens near minimally effective stimulation levels (which is already very well known). Obviously, the same critique applies to the current levels they use for the rest of the study. It would have been nice to see a more systematic and quantitative examination of the stimulus parameters where this phenomenon is strong versus weak.

We now show that the likelihood of observing an OFF response decreases with increasing stimulus amplitude in modified Figure 1. This dependence on stimulus amplitude is also seen in the slice data and is now shown in revised Figure 3.

2. Imaging from "single cells". Single cell imaging is not possible with epifluorescence illumination in a slice this thick, especially without neuropil removal. At best they are ROIs but even then, what is the value when there is no basis for the argument that their ROIs are independent response units? I suggest eliminating 2G, Figure 4 and Figure S4. The issues with Figure 4 (and S4) go way beyond the cellular resolution issues, they are uninterpretable for a number of reasons.

We have removed the section on deep learning, and therefore removed previous Figure S4. We respectfully suggest that single-photon imaging of calcium activity in slices has been done for some time (Maclean et al. 2005 PMID: 16337918, Berger et al. 2007 PMID: 17360827, Cameron et al. 2016 PMID: 27183102). We have now re-analyzed the data using neuropil correction as suggested by the reviewer and have found similar results and present them in modified Figures 2 and 5.

3. In Figure 2D, I'm not convinced that they are looking at signals from TRN cell bodies as opposed to thalamocortical/corticothalamic axons. The thalamic radiation forms a dense fascicle right at the point of the TRN in this plane so it could be very hard to distinguish between TRN cell bodies and axon signals with the optics they are using. Too bad they didn't express GCaMP in the GAD-Cre mice they use for inhibiting the feedforward inhibitory IC projection neurons.

While it is correct that corticothalamic axons pass through this area, we think that these are likely signals from the TRN for four reasons. First, flavoprotein imaging produces very weak signals in axons. Second, this region is the precise location of the auditory region of the TRN. Third, in the current studies, we show that application of a glutamate blocker to this region strongly facilitates thalamocortical transmission – a finding that is inconsistent with this region being reflective of axons of passage. Fourth, to definitively establish that there are TRN cell bodies in this region, we were able to record from several TRN cells, and found that their firing was synchronized with IC stimulation and that these neurons had current/voltage profiles consistent with TRN neurons. We added this result as Figure 2—figure supplement 3.

4. Calcium imaging in the MGB and IC. Looking at other publications of the Thy1-GCaMP6 mouse, I see lots of labeling in pyramidal neurons in neocortex and hippocampus, but no labeling in the MGB or IC. I'm confused about what is being measured in Figure 2E. Maybe there is expression in younger mice or is just very faint?

This is a good point. GCaMP signal is smaller (but present) in the IC and fairly weak in the MGB in the original description of these mice by Dana et al. (PMID: 25250714). Our data are consistent with their findings, with the smallest signals deriving from the MGB. It is also certainly possible that part of the signal that is observed in IC and MGB is contaminated by flavoprotein signals, which are obtained with the same filter set, but are generally about an order of magnitude smaller than GCaMP signals. Either way, it does not impact the main conclusion from the use of GCaMP mice which is that the cortical ON/OFF responses are not an artifact of metabolic imaging. We have modified the text accordingly.

5. Figure 3E-H (graphics could be greatly improved here) show that EPSCs in MGB are unchanged in cortical ON vs OFF trials but IPSCs are elevated in OFF trials. This is one of the mysteries of their study that they try to address with a bunch of uninterpretable analysis methods (Figure 4 and Figure S4). The conundrum they face is how does enhanced thalamic inhibition create a cortical OFF response without making the MGB excitatory response weaker? Figure 4 and S4 attempt to explain this but are contrived and unconvincing. I don't disagree but I would suggest that they are thinking about it too simplistically. It doesn't matter what the MGB cell bodies are doing, what matters is what they are transmitting to the cortex during OFF trials. It would be helpful if they quantified these data more carefully by looking at the relative timing of the IPSC and EPSC responses. Looking at the study by Reinhold, Lien and Scanziani 2015, the TRN can regulate sustained spiking and thalamocortical synaptic depression in ways that cannot be dismissed just by measuring the excitatory response. They don't have to fully answer this here. They can do the best with whatever straightforward analysis they can manage and then devote some space in the discussion to actually discussing alternative explanations for their experiments instead of rhapsodizing about how these data solve the greatest mysteries of the brain.

Thank you for these suggestions. We have modified the graphics on this figure. We have quantified the timing of the IPSCs and EPSCs in the modified version of figure 3 (now Figure 5). Also, we have discussed these issues in the revised Discussion section.

6. Figure 3K – It would be important to see the individual data points. The legend says the data are an n=5, reflecting 5 slices from 4 animals. How do you get two of these slices from one animal? Regardless, this is a very small sample to make such a strong conclusion as blocking cortical GABA_A_ receptors does not affect the OFF response probability. Was the AC Gabazine always performed after the MGB wash? The order should be counterbalanced and really I'm not even sure what the justification is for using a t-test on a sample this small. Unclear how the assumptions of the test can really be determined.

We are occasionally able to obtain two viable brain slices from a single animal retaining IC-MGB-AC connectivity. We have tested whether the data were normally distributed, and based on this result, ran a suitable non-parametric test. We have also re-run the pharmacology experiments in a counterbalanced fashion as suggested and obtained similar results (see modified Figure 4I-K).

7. I suggest putting all three of their halorhodopsin experiments into one figure (IC inhibitory feedforward neurons, L6 CT and L5 projection neurons). I'm also bothered that there isn't a control condition where mice have undergone a perinatal injection to express a control fluorophore. This seems particularly warranted because they injected an enormous volume of virus solution (1600 nL) into very small area. Apart from this control, while I think the L6 inactivation result is very interesting, I strongly disagree with their often repeated interpretation of the data, that "Population OFF responses were associated with TRN-mediated inhibition at the level of the MGB, under the control of layer 6 corticothalamic projections".

We appreciate the reviewer’s suggestion and we have now placed all of the halorhodopsin experiments into one figure. Also, we conducted a control experiment in which we injected 4 day old mice with the same serotype viral particles that do not express halorhodopsin receptor. We put the results of this experiment in revised Figure 6.

Their data do not support this conclusion. Yes, blocking AMPA receptors in TRN reduced the probability and Yes, silencing L6 CT also reduced the probability but it is a logical fallacy to assume those are causally related. Ntsr1 neurons make synaptic contacts on TRN neurons, MGB neurons, local AC inhibitory neurons and local AC excitatory neurons. It is entirely possible that the Ntsr1 neurons are mediating this effect through local connections in the cortex and not through the descending projection. In that regard, it is a bit frustrating that the main point of the Guo 2017 study was the Ntsr1 neurons can strongly bias AC neurons to become less responsive to afferent activity via intracortical connections, yet these findings are not mentioned in this context.Don't get me wrong, they might be right. But providing that the L6 CTs modify MGB via TRN would require a different type of experiment to isolate the influence of L6 CT axons onto TRN neurons and not the other three types of neurons that it communicates with. Again, good topic for a revamped Discussion section.

This is an excellent point that we had not considered previously. The reviewer is absolutely correct that silencing NTSR1+ neurons may have multiple effects outside of their projections to the TRN, particularly based on their intracortical projections. We have modified the Discussion accordingly.

8. Figure S3 should be upgraded to a main figure and quantified.

We appreciate the reviewer’s suggestion, and we upgraded the figure to a main figure 3 and quantified the results.

9. Figure S6 is interesting but preliminary. There are only four data points and DREADD experiments are usually considered uninterpretable without a control condition in which CNO is applied with the expression of a control fluorophore (i.e., without the designer receptor) to account for known off-target effects of CNO.

We appreciate the reviewer’s suggestion, and we have done the control experiment by injecting 4 days old mice with the same stereotype viral particles that do not express the DREADD receptor. We put the results of this experiment in Figure 6—figure supplement 1.

10. The issues with Figure S7 are that Rbp4-Cre does not selectively label "PT-like" L5 cells that project sub-cerebrally. In fact, most of their axons remain within the cortex. To this point, the expression they show in no way resembles the cartoon in Panel A but more problematically does not compare to the expression in the Ntsr1 neurons. As such, it is hard to interpret much from the negative result because it may be that they are hyperpolarizing many fewer neurons overall and very often not the L5 neurons that project to the MGB.

We agree with the reviewer’s comment. Initially, we conducted this experiment as a negative control thinking that inhibiting another cell type in the cortex, not layer 6 cells, will not be able to retrieve the OFF cortical response. We respectfully suggest keeping the RBP4 data in the paper to help differentiate layer 5 vs. layer 6 effects. Clearly if an effect was seen after RBP4 silencing, one could not know if the effect was due to subcortical vs. cortical effects.

11. Figure S8 is another example of them trying to squeeze too much out of their data. First of all, I was unclear about how they were confident that they patched Ntsr1 neurons and not other L6 cell types. But mainly the analysis methods were a little strange to me. Why not just filter in the Gamma band (or other bands)? Also there is a huge literature showing the phase of ongoing 2-6Hz oscillations gate the response probability of AC neurons (e.g., in Guo 2017). It would have been reasonable calculate the phase and amplitude of the pre-stimulus membrane currents from its analytical signal using the Hilbert transform. In it's current form, the quantification in this figure is not convincing at all.Apart from that, the Pre-stimulus L6 neuronal activity could only accurately predict 58.1 {plus minus} 3.5 SD% ON and OFF cortical responses, which is subtle compared to the theoretical random accuracy 55.1%. Moreover, should the random accuracy be calculated by training and testing the model with shuffled ON and OFF responses?

Based on the suggestion made by the reviewers and the editor, this part was removed.

Reviewer #3:This is an interesting paper that finds that the cortical feedback to the thalamic reticular contributes to the omission of responses in the cortex. This is an important and surprising finding, and the authors bring a whole arsenal of state-of-the-art tools to dissect the contribution of this pathway. The variety of the methods, that include calcium imaging, flavoprotein imaging, electrical stimulation, whole cell recording, pharmacologic/optogenetic/chemogenetic manipulations, all point towards the same conclusions. The data that are presented support most of the claims of the paper. Most of my concerns reflect the interpretation of the data, the focus of the introduction and discussion, some missing controls and the lack of report of statistics in the main body. I believe that the authors should be able to address these concerns with the data in hand, through additional analysis and re-interpretation.1. The emphasis in the introduction and discussion on the "classical models" of purely feedforward hierarchical processing seems somewhat forced. Numerous studies, especially in the auditory system, have addressed the importance of feedback in anatomy, auditory processing, learning and behavior. Rather than skipping over this important work, the paper would benefit from the discussion of previous work on the function of feedback in sensory processing. This would furthermore allow the authors to better define what role their work plays in this context.

Thank you for the suggestion. We have made substantial modifications to the Introduction in accordance with this suggestion.

2. Similarly, the emphasis on the role of this feedback in neuronal oscillations seems to be overstated. In the introduction, the discussion of hallucinations seems to come out of nowhere. In the paper, only one supplementary figure is devoted to analysis of oscillations, whereas in the abstract, introduction and discussion, these is an extensive focus on the gamma oscillations. The gamma activity-based prediction of ON vs OFF responses seems to be weak. The classifier only had an accuracy of 58%, which is not very high, and is only barely above random accuracy of 55. This focus and confusing result detracts from the main message of the paper.

Thank you for these suggestions. We have modified the Discussion with respect to the potential connection to hallucinations. Based on the suggestion made by the reviewers and the editor, the gamma oscillations has been removed.

3. The terminology of ON and OFF responses seems to be at odds with the accepted terminology in the auditory field. Typically, in the auditory field, ON responses refer to the activity at the onset of a stimulus, and OFF responses refer to the activity after stimulus offset. OFF responses, where there is no response at all, is confusing. The authors allude to the use of UP and DOWN states in the literature, but there is only indirect evidence that this activity indeed corresponds to those states. Perhaps using "Response" and "Omission"; yes/no; 1/0 or +/- could work.

Thank you and we have struggled with this terminology ourselves. Reviewing the classical auditory literature, most authors use “onset” and “offset” to refer to response at the beginning vs. the end of a tone rather than ON/OFF (Qin et al. 2007, PMID: 17360820, Anderson and Linden 2016, PMID: 26865621, Kopp-Scheinpflug et al. 2018, PMID: 30274606). We have been reluctant to call these UP/DOWN responses because we do not yet know how completely UP/DOWN aligns with ON/OFF. We thank the reviewer for the other suggestions, but also find them to be less descriptive. We respectfully suggest using the terms “population ON” and “population OFF” responses and have made the corresponding changes in the revised manuscript.

4. The claim that the thalamus selects co-activated cortical ensembles, referred to in the abstract, last sentence of the introduction as well as the discussion, is not supported by data. Perhaps it is known that the auditory cortex uses partially overlapping ensembles of coactivated neurons to represent auditory stimuli, but that data is not represented in this paper. In fact, the data in this paper hinges on an "all or nothing response," which undermines this idea of coactivated ensembles in the AC. These contrasting ideas deserve reconciliation.

We agree that our data do not establish that specific ensembles of neurons are activated by particular stimuli. We have replaced the word “ensemble” with “population” or “group” of neurons in the Abstract and Introduction, respectively, which we believe has less specific implications than “ensemble.”

5. Statistics for all claims should be reported in the main text. Some statistics are missing, for example, in Figure 5D, the distribution of the histograms.

Thank you, we have moved the statistics from the figure legend to the main text.

6. It is great that the paper starts with results in vivo. However, those are under anesthesia. Are OFF responses also present in awake animals, perhaps influenced by attention or arousal levels? Perhaps worth speculating in the discussion on the functional importance of this corticothalamic gating in awake behaving animals.

Thank you. We have added text to highlight this point in the revised Discussion.

7. The authors use a deep learning algorithm to establish the predictive power of the response, training on MGB data vs irrelevant striatum. How would the algorithm work for IC data? Furthermore, it is unclear whether deep learning is required for this analysis. Would more simple statistical measures, such as correlation or Granger causality measures work here as well? Can the predictive ability of layer 6 activity be improved with just using pre-stimulus layer 6 firing rate, as opposed to LFP power spectra?

We appreciate the reviewer’s suggestion but based on the suggestion made by the reviewers and the editor, this part was removed.

8. The authors observed cortical IPSCs during ON responses but not OFF responses. The importance of this finding would be better demonstrated by comparing it with cortical EPSCs. Presumably, ON responses are characterized by large EPSPs and medium IPSCs, whereas OFF responses are characterized by no EPSCs or IPSCs. That way, it's clearer that OFF responses are not simply caused by increase in IPSC magnitude. It could then be contrasted that with the increase in IPSC magnitude observed in MGB, as they show in figure 3G. The importance of figure 3E/3F is unclear. It would be great to explain why IPSCs are observed in cortical ON responses and not OFF responses.

We apologize for the lack of clarity. The important outcome of these experiments is the absence of cortical IPSCs during OFF responses. The presence of barrages of IPSCs during ON responses was initially surprising to us, but is consistent with ON responses being found during UP states. UP states are associated with barrages of both EPSCs and IPSCs (Hasenstaub et al. 2005, PMID: 16055065, Tahvildari et al. 2012, PMID: 22933799, Salkoff et al. 2015, PMID: 26180200). We have now quantified the magnitude of the IPSCs during ON vs. OFF responses (revised Figure 4F) to make this point more clear, and to provide more contrast with the MGB IPSC magnitudes, which were larger during the OFF responses.

9. How well do gabazine and NBQX stay localized to the area of injection? Is there any chance of diffusion of the agent confounding results? How was localization confirmed?

We mixed the drug and the ACSF as control with Alexa Fluor-594 dye, so we could visualize the injection of the solution injected. We provided the images or the videos of these injections. Figure 4—figure supplement 1 shows the localized injection of the drugs.

10. What are the controls for chemogenetics?

Thank you for this suggestion. We have now done the control for chemogenetic probe and we put the results in Figure 6—figure supplement 1.

11. In Figure 4, when performing calcium imaging in MGB, which trials had cortical ON vs OFF responses? Was that determined through simultaneous LFP recording in AC (diagram in figure 4A)?

Yes, there was a simultaneous LFP recoding and MGB calcium imaging. We have modified the text to make this point more clear.

12. How well can latency of response and response timing variability (Figure 4) be assessed using GCaMP6s, which has a response time scale on the order of seconds? Is the variability in response timing really on the order of hundreds of milliseconds (Figure 4D)? Perhaps this would be better evaluated using whole cell recording or electrophysiology.

We agree that there are significant limitations in the interpretability of response timing using calcium signals. However, gold-standard approaches to measuring timing, such as electrophysiology in the slice, limit the number of neurons simultaneously sampled. We have now added text describing the limitations to this approach.

13. Is it MGB synchrony, or MGB inhibition, that leads to cortical ON vs OFF responses? The importance of MGB synchrony, or at least its interaction with TRN-MGB inhibition, is unclear. Furthermore, the synchrony analysis should be conducted with more appropriate statistical tools, including correlation measures. It remains unclear how a desynchronized MGB leads to an OFF response (aka no firing) in the cortex. Do layer 4 cortical cells not reach firing threshold when the inputs are asynchronous? Or do cortical neurons not receive inputs from MGB when the neurons fire asynchronously (due to further gating by third party cells)?

We believe that the data suggest that inhibition of the MGB leads to desynchronized MGB output, leading to OFF responses. Synchrony in this case was measured as variance in latency across the population which is a direct measurement of whether MGB neurons are responding at the same time. The idea that thalamic synchrony is needed to elicit a cortical response and that TRN may disrupt thalamic synchrony is established in the literature (Bruno and Sakmann et al. 2006, Pita-Almenar et al. 2014 PMID: 25339757). We have made this point more clearly in the revised Discussion.

14. Is the difference in power at ~36Hz significant when controlling for multiple comparisons across 11 different LFP power spectra frequencies? Perhaps the authors could try running a 2-way ANOVA with frequency and ON/OFF status as factors.

Based on the suggestion made by the reviewers and the editor, the gamma oscillations part was removed.

15. I recommend tightening the writing style throughout: e.g. in abstract: "Here, we elucidate the mechanism for gating of population activity." Some paragraphs are multiple pages long. There are run-on sentences, and the authors use passive tense extensively. I would recommend splitting up the paragraphs, so that each paragraph contains one result/analysis, as well as two sentences for motivation and conclusion.

We appreciate the reviewer’s suggestion and have revised the manuscript extensively.

[Editors' note: further revisions were suggested prior to acceptance, as described below.]

The authors are to be commended for carrying out a number of new experiments to provide essential control data and to provide additional information that has strengthened the study. The majority of the other concerns raised by the reviewers have been dealt with adequately, but there are a few remaining issues that need to be addressed.Both the in vivo and slice data now clearly illustrate the dependence of the cortical responses on the stimulus magnitude, with the probability of an OFF response occurring declining with increasing stimulus strength. This reinforces one of the points made by reviewer 2, namely that these effects appear be present only for near minimally effective stimulation levels. That raises the question of what function they serve and how these responses contribute to our understanding of cortical processing under more natural conditions (including in awake animals at higher sound levels). These are important issues that need to be addressed in the Discussion.

We agree and have added the following text to the Discussion:

“Moreover, the strong association between the high probability of the OFF cortical responses and the low level of stimulation suggests that cortico-reticulothalamic control of thalamocortical transmission is most likely to be effective when signal-to-noise ratio is low. This finding is consistent with the notion that top-down modulation is mostly required for the attentional modulation of weak signals and that highly salient signals rely on bottom-up mechanisms to activate perceptual representations (reviewed in [45]).”

The readability of the text would benefit further from greater use of paragraphs (as pointed out by reviewer 3). For example, one paragraph runs from lines 304-380. This does include some deleted text, but is still far too long.

Thank you for the suggestion. We have broken this paragraph up into two paragraphs. The second paragraph focuses on the TRN experiments only.

Line 364: 5J should be 6J.

We have corrected this error.

Line 423: "exposure to pure tones" might be construed as a form of passive acoustic environment. "Presentation of pure tones" would be less ambiguous.

We have changed this phrase as suggested

Figure 1 legend: there are several full stops missing.

We have added the full stops.

Figure 2 – Supplementary Figure 3: the legend refers to red and green boxes, but there are none in the figure (at least in the composite pdf of the manuscript).

Thank you for finding this error. These colors refer to the previous version of the manuscript. We have changed the legend.

Line 943: the sentence ends "showing".

Thank you. We deleted the word “showing.”

Figure 6 – Supplementary Figure 1G: all traces are black, whereas the legend claims that some are brown or green.

Thank you. We removed references to brown/green in the legend.